# A Balanced Neuro-Symbolic Approach for Commonsense Abductive Logic

**Joseph Cotnareanu** [1, 3, 4]**, Didier Chetelat**[2]**, Yingxue Zhang**[2]**, Mark Coates**[1,3,4]
[1] McGill University [2] Huawei Noah's Ark Lab
[3] International Laboratory on Learning Systems (ILLS) [4] MILA
{joseph.cotnareanu, mark.coates}@mail.mcgill.ca,
{didier.chetelat, yingxue.zhang}@huawei.com

## Abstract

Although Large Language Models (LLMs) have demonstrated impressive formal reasoning abilities, they often break down when problems require complex proof planning. One promising approach for improving LLM reasoning abilities involves translating problems into formal logic and using a logic solver. Although off-the-shelf logic solvers are in principle substantially more efficient than LLMs at logical reasoning, they assume that all relevant facts are provided in a question and are unable to deal with missing commonsense relations. In this work, we propose a novel method that uses feedback from the logic solver to augment a logic problem with commonsense relations provided by the LLM, in an iterative manner. This involves a search procedure through potential commonsense assumptions to maximize the chance of finding useful facts while keeping cost tractable. On a collection of pure-logical reasoning datasets, from which some commonsense information has been removed, our method consistently achieves considerable improvements over existing techniques, demonstrating the value in balancing neural and symbolic elements when working in human contexts.

## 1 Introduction

Large Language Models (LLMs) have demonstrated impressive abilities to reason formally, often via chain-of-thought reasoning (Wei et al., 2022). While the state of the art modern LLM-based systems show impressive reasoning capabilities, it is unclear whether this comes from the LLM itself, or sophisticated post-learning refinement algorithms. At the same time, open-sourced LLMs still demonstrate an inability to naturally scale to problems that require complex proof planning (Saparov and He, 2023; Dziri et al., 2023). Such problems are exactly the type on which symbolic logical solvers excel: such solvers have a long history and were for a long time considered a key component of any path to artificial intelligence (Nilsson, 1991). Nevertheless, they are greatly restricted by their need for problems to be stated in symbolic language and for every relevant fact to be provided as input. These constraints have ultimately limited them to highly specialized applications, and they have never had the broad impact that was hoped for (Crevier, 1993).

These complimentary strengths of neural and symbolic methods have motivated a revival of interest in neuro-symbolic methods, where an LLM incorporates a logic solver to improve its reasoning abilities (Ye et al., 2023; Lee and Hwang, 2024; Lyu et al., 2023; Olausson et al., 2023). In these approaches, the LLM translates problems formulated in natural language into symbolic language, addressing one of the key deficiencies of a purely symbolic approach. Nonetheless, these hybrid systems remain impractical because they are ultimately purely deductive: that is, every relevant fact must be provided as input. This means that the symbolic solvers are often unable to reach a conclusion simply because obvious, commonsense assumptions are left unstated, and it is often difficult to predict which should be included until one is presented with a failed reasoning chain.

For example, consider the problem in Figure 1. A logic solver would return "unknown" for the target query as, formally speaking, neither its truth nor its falsehood is implied by the premises. A human, however, would easily solve this problem by supplying the additional commonsense fact that white

Context: Some animals tough winter out. They do not leave. They do not hide. They must survive. Sometimes nature helps them out. Some animals grow thicker coats in the winter. Other animals change color. The arctic fox is brown in the summer. His coat turns white in the winter.

Question: The arctic fox's coat turns white in the winter because white absorbs the sun and is warmer.

$\exists x : tough\_out(x, winter)$
$\exists x : \neg hide(x) \wedge \neg leave(x) \wedge survive(x)$
$\exists x : helps(nature, x)$
$\exists x : grow\_coat(x, winter)$
$\exists x : change\_color(x, winter)$
$brown(fox, summer)$
$turns\_white(fox, winter)$

$absorbs(white, sun)$
$warmer(coat)$
$turns\_white(fox, winter)$

Figure 1: An example from a children's comprehension exercise booklet [1]. Left: the problem phrased in human language. Right: the same problem translated to first-order-logic.

surfaces reflect light ($turns\_white(fox, winter) \rightarrow reflects(fox, sun)$). This ability to supply missing information is usually themed abductive reasoning, and is a key mark of human intelligence.

The limitation of current neuro-symbolic LLM systems to deductive reasoning means that they have mostly been so far of theoretical interest, since they tend to break down when confronted with more complex problems where enumerating every possible background fact is not realistic. However, besides their translation skills, LLMs possess also another striking ability: their training on prodigious amounts of internet data has made them very adept at recognizing commonsense statements, to the point where they have been regarded as potential universal databases (Petroni et al., 2019). In a way, LLMs seem to have internalized most commonsense knowledge.

This realization has led some works to use an LLM itself to supply missing but relevant clauses when reasoning. Notably, Toroghi et al. (2024) proposed a method that operates an exhaustive search over a heavily restrained set of rules in the symbolic space, whereas Liu et al. (2024) proposed a method that uses LLM prompting to produce new rules which might be deducible from the given logical context. While these methods lie on opposite ends of the symbolic-linguistic reasoning spectrum (Figure 2), they both limit themselves to searching over such a restricted space of possible commonsense that they cannot solve practical problems.

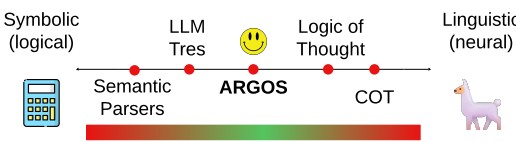

Figure 2: Symbolic-Linguistic Spectrum depicting the positioning of LLM-Tres (Toroghi et al., 2024), Logic-of-Thought (Liu et al., 2024), and Chain-of-Thought (COT) (Wei et al., 2022) relative to our approach.

In this work, we seek to improve AI reasoning abilities by using an LLM to provide relevant unstated commonsense clauses to a logic solver, but unlike previous works, without imposing significant constraints on the shape or content of such clauses. Furthermore, and most importantly, our method ARGOS (**A**bductive **R**easoning with **G**eneralization **O**ver **S**ymbolics) can abduce propositions not previously instantiated in the input problem. To compensate for the far more general search space, we guide the search using feedback from the logic solver in the form of the SAT problem backbone, another novel contribution. The resulting system strikes a balance between linguistic and symbolic approaches, allowing us to use both their strengths while minimizing their weaknesses to achieve true abductive reasoning.

The contributions of this paper are as follows.

- We propose a novel framing of the commonsense logical reasoning problem founded upon classical logical principles and an aim towards more realistic use-cases.

---

[1]Taken from `https://www.ereadingworksheets.com/worksheets/reading/nonfiction-passages/wintertime`. We selected a choice from multiple choice question 3 and re-phrased it as a True/False question, according to the logic-problem framing.

- We introduce a novel algorithm that (i) searches over larger spaces of commonsense facts; and (2) uses logic solver feedback in the form of the backbone graph to increase practicality and efficiency.

- We demonstrate empirically on multiple benchmarks and large language models that our method improves substantially over existing symbolic and neural methods on abductive reasoning problems where background information is missing.

## 2 RELATED WORK

Previous LLM-related logical reasoning methods combine symbolic and neural approaches, but usually rely much more on one or the other. Appendix G provides an extended review.

**Neural Methods**   Wei et al. (2022) were the first to present a framework for LLM-based reasoning, showing that providing examples of rationales for answers to questions can induce the LLM to do the same, leading to improved accuracy. Kojima et al. (2022) showed that this can be induced without any few-shot examples by prepending the sentence "Let's think step by step" before generating an answer. This is known as "Chain of Thought" (COT). Following this, Wang et al. (2023) proposed self-consistency (SC), using COT multiple times and taking the mode as the prediction. However, Saparov and He (2023) observed that COT and SC suffer from challenges in proof planning — rationale steps tend to be factual but of low value. This motivated guidance of the LLM at a step-level. Yao et al. (2023) proposed Tree of Thoughts (TOT), which explores hand-crafted trees using an LLM to solve reasoning tasks. TOT is poorly suited to logical reasoning settings as logic problems have highly variable tree-structures. Kazemi et al. (2023) and Lee and Hwang (2024) proposed more logic-focused methods, with reverse reasoning, starting at the answer and ending at the problem. These back-chaining methods, however, underperform symbolic approaches.

**Symbolic Methods**   Acknowledging that LLMs are poor proof-planners, a series of methods, including F-COT (Lyu et al., 2023) and SAT-LM (Ye et al., 2023), proposed to offload the reasoning burden from the LLM to more specialized tools. In these works, the LLM converts the text to symbolic logic, and a solver is then employed. Logic-LM (Pan et al., 2023) extended this to include a self-refinement step. While these methods perform well on simple datasets, they fail to account for ambiguity and the exclusion of common knowledge. Addressing this, Liu et al. (2024) and Wang et al. (2022) proposed algorithms that produce new clauses via logical deduction and then add the logic back to the text for an LLM to solve. While this might help the LLM, it does not add information to the problem, because any added relations are already deducible. Instead of producing clauses via deduction, Toroghi et al. (2024) proposed a method that exhaustively searches for new single-proposition modus-ponens clauses. However, the search is conducted only over the propositions from the question, and repeated until the problem is solvable by classical logic, diminishing robustness. This search space is highly restricted and leaves out nearly all necessary information for some logic problems.

## 3 BACKGROUND

**Propositional logic** is a logical system built around propositions, which are statements of fact such as "It is sunny" or "I need an umbrella" which can be true or false. Propositions are often denoted by single letter variables such as $A$ or $B$, called a propositional variable, which can be tied together by logical connectives (such as $\wedge$, $\vee$ or $\rightarrow$) to form further compound propositions.

A **deductive propositional logic problem** is composed of a set of propositional variables $\mathcal{V}$, a set of propositions (each represented by a propositional variable in $\mathcal{V}$) and compound propositions (built by using logical connectives to connect propositions by their representative propositional variables in $\mathcal{V}$) called the premises $\mathcal{P} = \{P_1, \ldots, P_K\}$, and a proposition or compound proposition $Q$ also built from those variables, called the query. The premises are given to be true ($\vdash \mathcal{P}$), and the goal of the problem is to determine whether they imply the query, $\mathcal{P} \vdash Q$, or its negation, $\mathcal{P} \vdash \neg Q$. Such problems are usually solved by translating them into two Boolean Satisfiability (SAT) problems, one for $Q$ and one for $\neg Q$. Let $\mathcal{L}(\mathcal{V}) = \{A \mid A \in \mathcal{V}\} \cup \{\neg A \mid A \in \mathcal{V}\}$ denote the set of all so-called literals of the problem. The **backbone** of the problem is the collection of all those literals which are implied by the premises, backbone$(\mathcal{P}) = \{L \in \mathcal{L} \mid \mathcal{P} \vdash L\}$. In effect, they are values for the propositions represented by the variables in the problem which can be inferred from the premises. In

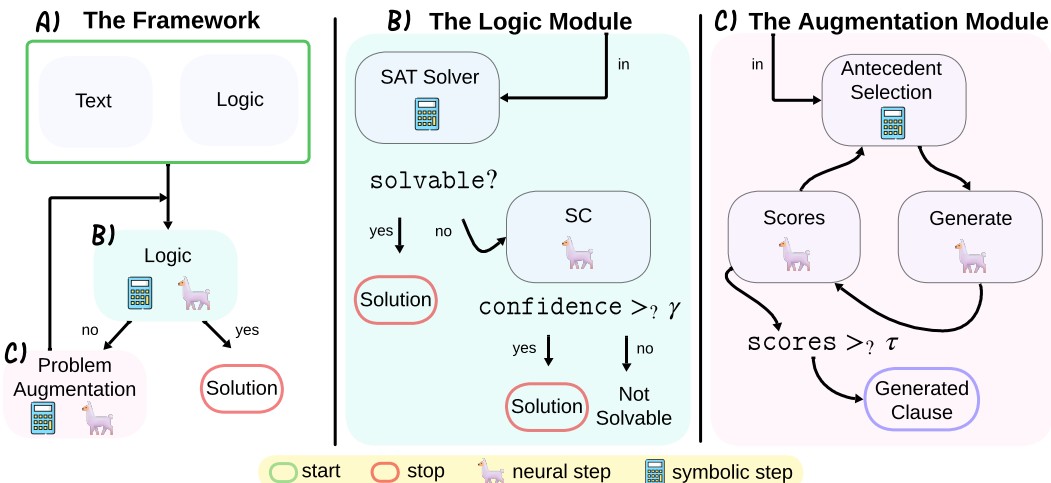

Figure 3: ARGOS at a glance. See Section 4.1 and Appendix F for details. (A) Given a propositional logic problem, we iteratively augment the problem with new propositions until it is solvable. (B) We attempt to solve the problem both with a logic solver, and with self-consistency (Wang et al., 2023). (C) If we fail, we attempt to add additional commonsense propositions by combining literals from the backbone as antecedents, and generating a right-hand-side using an LLM. We test the proposition for commonsense and relevance using this same LLM, and add it to the pool if it passes the tests.

an **abductive commonsense propositional logic problem** the premises $\mathcal{P}$ entail neither the query $Q$ nor its negation $\neg Q$: the problem is underdetermined. Instead, one must augment the premises with additional commonsense propositions $\mathcal{C}$, which represent background facts or knowledge left unstated in the problem, until either $(\mathcal{P} \wedge \mathcal{C}) \vdash Q$ or $(\mathcal{P} \wedge \mathcal{C}) \vdash \neg Q$. Thus, the goal of an abductive problem is to not only find the truth-value of $Q$, but also a corresponding set of commonsense propositions $\mathcal{C}$ to complete the problem. We assume that $\mathcal{P} \wedge \mathcal{C} \not\vdash \bot$, that is that the premises $\mathcal{P}$ are not contradictory with commonsense (i.e. that $P$ and $C$ are consistent). One can show that, under this assumption, the answer to the problem will not depend on the choice of commonsense set $\mathcal{C}$: details are provided in Appendix A.

In practice, the problems we encounter in real life are often stated in terms of first-order logic. **First-order logic** is a logical system that extends propositional logic to entities and their predicates. An $n$-ary predicate is a symbol of a relation, such as $MotherOf$, that takes as arguments $n$ terms such as $x$ and $y$ to become a formula $MotherOf(x, y)$, and becomes true or false when constants, such as $Alice$ and $Bob$, are used as a grounding for its arguments. Predicates can be connected by logical connectors, and can also be quantified over a discrete or abstract set of entities with $\forall$ and $\exists$, to form compound propositions such as $\forall x \forall y [MotherOf(x, y) \rightarrow \neg Male(x)]$.

First-order logic formulas over a finite set of entities can always be converted into equivalent propositional logic formulas, a process known as **grounding**, by instantiating a propositional variables for every predicate $F(x)$ and entity $A$, and expanding $\forall x \, F(x)$ into the compound proposition $(F(A) \wedge F(B) \wedge \dots)$ and $\exists x \, F(x)$ into $(F(A) \vee F(B) \vee \dots)$. Given two propositional literals, we will declare them **related in first-order logic** if they have an entity in common. For example, $MotherOf(Alice, Bob)$ and $\neg Male(Alice)$ are related because both involve the entity $Alice$.

### 3.1 PROBLEM STATEMENT

We are given an abductive propositional logic problem in both textual and logical form, as defined in Section 3, and we are also provided with a large language model and a SAT solver. As described, the task is to determine whether the target query is true or false given the premises and some additional commonsense propositions which must be found. Four annotated examples are provided, intended for few-shot prompting. In particular, the task is inference-only and no training phase is involved. We evaluate performance based on the number of correctly answered questions on a test dataset.

## 4 METHODOLOGY

We now describe our novel algorithm to tackle the problem described in Section 3.1. This algorithm is described by the diagram in Figure 3, and formally as Algorithm 1 in Appendix D.

### 4.1 ALGORITHM

We start the algorithm by initializing our set of commonsense propositions as empty, $\mathcal{C} = \{\}$. As shown in module B of Figure 3, we first try to solve the problem using the SAT Solver (`sat_solve`) to test whether either $(\mathcal{P} \wedge \mathcal{C}) \vdash Q$ or $(\mathcal{P} \wedge \mathcal{C}) \vdash \neg Q$. If it reaches one of these conclusions, our job is finished; if not, we at least obtain from our call the backbone $\mathcal{B} = \{L \in \mathcal{L} \,|\, \mathcal{P} \vdash L\}$.

Next, still in module B of Figure 3, we attempt to solve the problem using the LLM (`llm_solve`) by $k$-shot self-consistency (we use $k = 5$ in our experiments). We ask the LLM whether the query is true or false, providing it the premises and the commonsense found so far. Details can be found in Appendix F.1. If the fraction of votes pass a certain threshold $\gamma$, we also conclude either $(\mathcal{P} \wedge \mathcal{C}) \vdash Q$ or $(\mathcal{P} \wedge \mathcal{C}) \vdash \neg Q$ respectively, and the algorithm is finished. This parameter $\gamma$ (initialized at $\gamma = 1$ in our experiments) is reduced by a fixed amount $\gamma \leftarrow \gamma - \alpha$ at every iteration ($\alpha = 0.1$ in our experiments). Thus, the maximum cost of our algorithm, in terms of number of COTs required, is bounded at $cost < k\frac{\gamma-0.5}{\alpha}$, since when $\gamma = 0.5$, the fraction of votes is guaranteed to pass the threshold since the vote is over binary classes. For details on empirical cost, see Appendix B.

If neither solving method succeeds in establishing $Q$ or $\neg Q$, we try to add a new commonsense proposition to our pool $\mathcal{C}$, as illustrated in module C of Figure 3. In practice, we define a proposition to be commonsense if it seems true to a large language model without any context. To guarantee that the added proposition will grow the problem's backbone, we search for commonsense propositions of the form $L_1 \wedge L_2 \rightarrow L_{\text{right}}$, where $L_1$ and $L_2$ are literals in the backbone $\mathcal{B}$, and $L_{\text{right}}$ is a new literal suggested by the LLM. Note that $L_1$ and $L_2$ may be the same literal, in which case we in effect have a formula of the form $L_1 \rightarrow L_{right}$, thereby allowing both single and two-literal antecedents. In addition, by adding $\emptyset$ to the set of backbone literals, we can also have $\emptyset \rightarrow L_{right}$, allowing 0-literal antecedents. This search routine (`find_new_commonsense`) is described in Algorithm 2 in the Appendix. In detail, we start by iterating over pairs of literals in the backbone. We iterate by prioritizing the literals that share the most entities with others in the backbone, $\text{score}_{\mathcal{B}}(L) = \#\{L' \in \mathcal{B} \,|\, L' \text{ has an entity in common with } L\}$, so that we take highly-scored literals first. This gives a measure of relevance of the literal to the problem. To understand the rationale behind this choice, consider an example in which six relations are known about John and only one is known about Jane. If asked to guess about whom the problem is about, the natural guess would be John, since while problems often include extraneous information, it is rare that the majority of the problem is extraneously included. Next, for a given pair of literals $L_1, L_2$, we prompt the LLM (`llm_generate`) to generate a right-hand-side literal $L_{\text{right}}$ for $L_1 \wedge L_2 \rightarrow L_{\text{right}}$. In doing so, the LLM might introduce new variables not previously involved in the problem. Details can be found in Appendix F.2. We choose this forward-chaining approach rather than a goal-oriented backwards-chaining for simplicity, since LLMs are much easier to prompt for forward-chaining (COT) than backwards chaining (recursive algorithms such as LAMBADA).

Finally, for each generated $L_{\text{right}}$, we use the LLM (`llm_score`) twice to evaluate it. First, we use the LLM (`llm_commonsense_score`) to score whether $L_1 \wedge L_2 \rightarrow L_{\text{right}}$ is likely to be commonsense. Second, we use the LLM again (`llm_relevance_score`) to score whether $L_1 \wedge L_2 \rightarrow L_{\text{right}}$ is likely to be relevant to our current context. Each procedure returns a probability between 0 and 1. Details can be found in Appendices F.3 and F.4, respectively, and human evaluation in F.5.

We stop the search at the first new proposition $L_1 \wedge L_2 \rightarrow L_{\text{right}}$ whose commonsense and relevance scores are both above a given threshold $\tau$ (we use $\tau = 0.3$ in our experiments). When this happens, we update the commonsense set $\mathcal{C}$ with this new proposition, and restart the process. If not, running new iterations will not change anything and we fall back on our best guess, namely the self-consistency estimate. In addition, if after multiple iterations the self-consistency threshold reaches zero, we also exit with the self-consistency estimate.

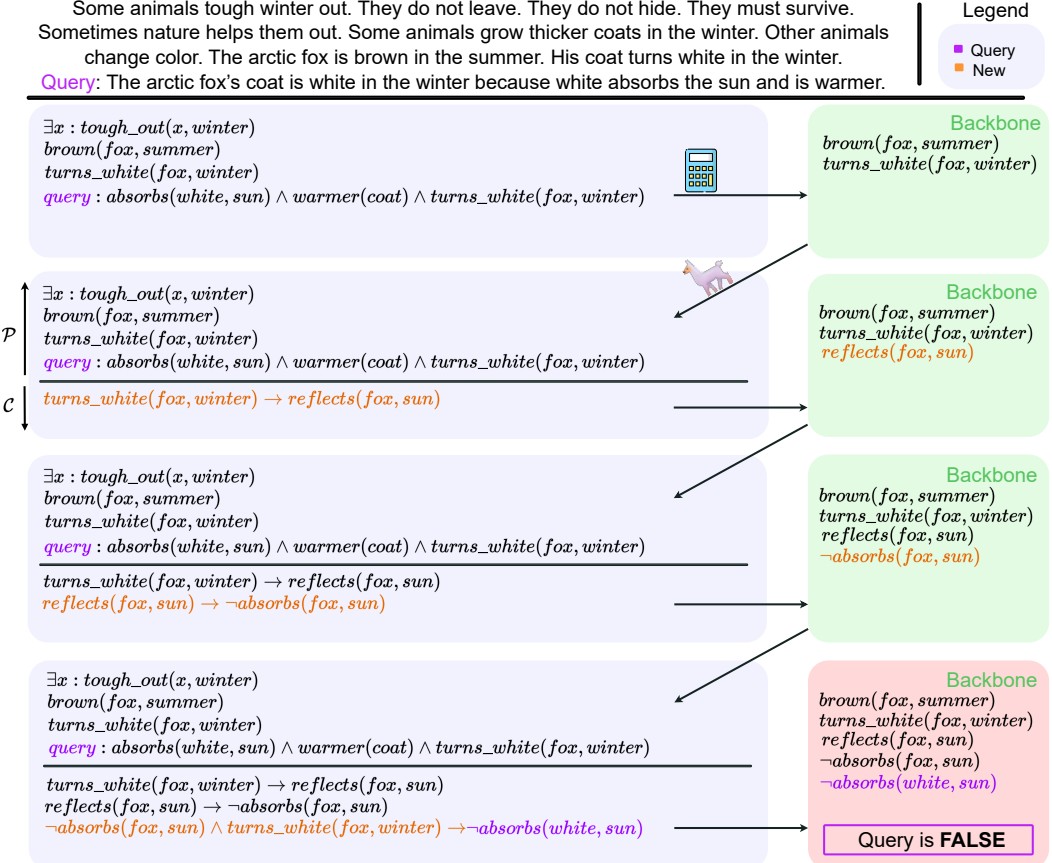

Figure 4: Overview of ARGOS with the winter fox example. We iteratively add to the logic problem and query a logic solver to look for conflicts within the backbone compared to the query. Eventually, we find that $absorbs(white, sun)$ is $False$, contradicting the query.

## 4.2 EXAMPLE

Consider again the winter fox problem from the introduction section. Let us describe in Figure 4 a hypothetical run of our ARGOS algorithm to illustrate how it could solve the problem. To simplify the illustration, let us use only the SAT solver, and not self-consistency. We start with the premises (in black) and the query (in purple) on the top left-hand-side.

We first run the logic solver, which fails to reach any conclusion, but returns an initial backbone. The algorithm chooses the antecedents $L_1 = L_2 = turns\_white(fox, winter)$ from this backbone, generating a new proposition $turns\_white(fox, winter) \rightarrow reflects(fox, sun)$. It is commonsensical and relevant to the question, so we add it to the question. We call the SAT solver again, which adds $reflects(fox, sun)$ to the backbone. Next, the algorithm selects the antecedent $L_1 = L_2 = reflects(fox, sun)$ from the new backbone and generates $reflects(fox, sun) \rightarrow \neg absorbs(fox, sun)$, which is similarly commonsensical and relevant. The SAT solver is called again and adds $\neg absorbs(fox, sun)$ to the backbone. Finally, in the third iteration ARGOS picks $L_1 = \neg absorbs(fox, sun)$ and $L_2 = turns\_white(fox, winter)$ from the backbone and generates $\neg absorbs(fox, sun) \land turns\_white(fox, winter) \rightarrow \neg absorbs(white, sun)$, which is a logical conclusion it deems consistent with commonsense and relevant to the question. At this point, we call the SAT solver again, which concludes that $\neg absorbs(white, sun)$ is true and therefore that the query must be false, returning $\mathcal{P} \land \mathcal{C} \vdash \neg Q$ as conclusion.

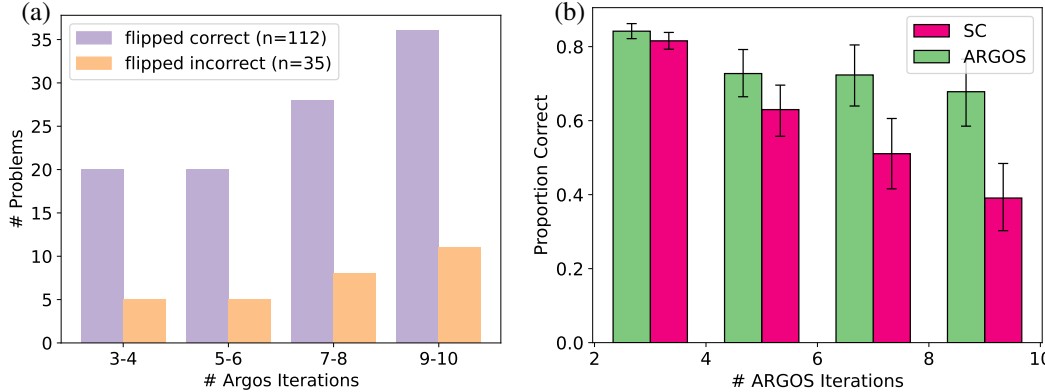

Figure 5: (a) The number of CLUTRR problems for which ARGOS flips SC predictions correctly and incorrectly. (b) SC and ARGOS accuracies on CLUTRR subsets, partitioned by the number of ARGOS iterations each datapoint receives.

## 5 EXPERIMENTS

**Models** We employ Llama3-8B (L8B), Llama3-70B (L70B), and Mistral 7B (M7B) as LLMs. Our method is dependent on access to logit-level outputs, so closed-source models are excluded. [2]

**Benchmarks** Unfortunately, there are few natural language reasoning datasets that are strongly logically-structured *and* commonsense-abductive. However, given a dataset of classical commonsense-based logic problems, data transformations to introduce the need for abductions are typically achievable. For a list of common datasets which have proven unsuitable for our setting, and corresponding explanation, see Appendix I. For our experiments, we use abductive versions of ProntoQA (Saparov and He, 2023), CLUTRR (Sinha et al., 2019), and FOLIO (Han et al., 2024). CLUTRR is not originally True/False, but it is multiple-choice. We modify it to be True/False output by making the question randomly either ask if the correct or an incorrect choice is True. While these datasets are better described with first-order logic, we render them propositional by unrolling their quantified formulas over all instantiated terms. While strongly logical and therefore obvious choices, these datasets are not representative of real-world application or generalizability of our method. To test our method's generalizability as well as its broader applicability to real use-cases, we also include some datasets that are not strictly logical. CosmosQA (Huang et al., 2019) and QUAIL (Rogers et al., 2020) are reading comprehension MCQA datasets. Reading comprehension is key for general summarization and interactive QA tasks, which are certainly a common LLM use-case in practice. ESNLI (Camburu et al., 2018) is a short-form natural-language-inference dataset. Each of these datasets requires some form of reasoning, but the structure of both the text and the necessary reasoning is generally fuzzy, requiring subjective interpretation. For the MCQA datasets, we process them into True/False questions similarly to how it was done for CLUTRR. We note that ProntoQA, CosmosQA and ESNLI performances are already saturated by self-consistency. Despite this, the results are valuable as they demonstrate that on these apparently simple tasks ARGOS is able to compare with purely neural methods, avoiding the performance collapse that more symbolic methods encounter. For few-shot examples, we randomly remove four problems from each dataset and annotate them with COTs, using the same four examples and annotations for each method. For more dataset details, please see Appendix H.

**Evaluation** We compare against *COT* (Wei et al., 2022), *Self-Consistency* Wang et al. (2023), *SAT-LM* (Ye et al., 2023), *Logic-of-Thoughts* (Liu et al., 2024) and *LLM-Tres* (Toroghi et al., 2024). For a fair comparison with 20-shot self-consistency and LOT, we set ARGOS's hyperparameters such that it makes no more than 20 COT calls per problem on average. Details are provided in Appendix B. We report accuracy on the abduction-modified evaluation sets and report results in Table 1.

---

[2]Experiments are each conducted on 1 or 2 NVIDIA Tesla V100 GPUs, depending on the LLM's GPU memory requirement. As a logic solver, we use Cadical (Biere et al., 2024).

## 5.1 RESULTS AND DISCUSSION

As can be seen in Table 1, ARGOS provides significant performance improvements over existing methods (up to +13%). Of the datasets, FOLIO is the most representative of human-generated logical reasoning problems. ARGOS outperforms the baselines for FOLIO, improving performance by 3-10%. For more structured problems (CLUTRR), the symbolic components of ARGOS become more reliable, and we see more consistent gains of 6-8%. On QUAIL, a highly ambiguous dataset that is also formatted in strange ways due to it being constructed by scraping forums and wikis, ARGOS improves compared to self-consistency by up to 13%, demonstrating its ability to adapt to even non-logical contexts. On ProntoQA, ESNLI and CosmosQA, despite the very competitive neural baseline performances, ARGOS performs comparably. Symbolic baselines (SAT-LM, LoT-20, LLM-Tres) see large performance gaps, at times being reduced to guessing. SAT-LM, despite the fact that some datasets are strongly logically structured and that we filter out mis-translated problems, still can not answer problems. Even in the best case, it is impossible for purely symbolic methods to handle realistic reasoning scenarios. LLM-Tres, despite having abductive capabilities, is so restricted in its abduction space that it it almost never capable of identifying the necessary rules to solve CLUTRR or ESNLI problems.

Table 1: Binary classification accuracy (True/False) of all methods on the datasets, using the chosen language models. Bolded text indicates that the method has the best performance, and that its performance is better than the next-best-performing method in a statistically significant way ($p$-value < 0.005 according to a Wilcoxon pair-wise rank test). Small-font numbers to the right indicate the bounds of the 95% confidence interval, derived via a bootstrap approach.

| | FOLIO | | | CLUTRR | | | PQA | | |
|---|---|---|---|---|---|---|---|---|---|
| | M7B | L8B | L70B | M7B | L8B | L70B | M7B | L8B | L70B |
| SC20 | 66% $^{66.4}_{65.5}$ | 71% $^{71.7}_{70.1}$ | 77% $^{77.7}_{75.9}$ | 59% $^{59.3}_{58.8}$ | 69% $^{69.5}_{68.8}$ | 69% $^{69.4}_{68.8}$ | 97% $^{97.2}_{95.6}$ | 95% $^{95.6}_{94.4}$ | 93% $^{94.1}_{92.4}$ |
| COT | 66% $^{66.4}_{65.5}$ | 68% $^{69.1}_{67.2}$ | 72% $^{72.5}_{71.8}$ | 59% $^{59.3}_{58.8}$ | 68% $^{68.4}_{67.8}$ | 66% $^{66.3}_{65.6}$ | 82% $^{82.9}_{81.7}$ | 90% $^{91.2}_{89.6}$ | 93% $^{94.1}_{92.4}$ |
| SAT-LM | 43% $^{43.2}_{42.8}$ | 43% $^{43.2}_{42.8}$ | 43% $^{43.2}_{42.8}$ | 50% $^{50.4}_{49.9}$ | 50% $^{50.4}_{49.9}$ | 50% $^{50.4}_{49.9}$ | 50% $^{50.3}_{49.8}$ | 50% $^{50.3}_{49.8}$ | 50% $^{50.3}_{49.8}$ |
| LoT-20 | 57% $^{57.3}_{56.6}$ | 69% $^{69.5}_{68.7}$ | 70% $^{70.4}_{69.5}$ | 71% $^{71.6}_{70.7}$ | 70% $^{70.2}_{69.7}$ | 69% $^{69.3}_{68.7}$ | 88% $^{88.4}_{87.5}$ | 97% $^{98.2}_{96.3}$ | 95% $^{95.7}_{94.3}$ |
| LLM-Tres | 66% $^{66.6}_{65.9}$ | 63% $^{63.2}_{62.4}$ | 63% $^{63.2}_{62.4}$ | 51% $^{51.5}_{50.8}$ | 51% $^{51.6}_{50.8}$ | 53% $^{53.2}_{52.8}$ | 80% $^{81.4}_{79.2}$ | 83% $^{83.8}_{82.3}$ | 76% $^{76.6}_{75.2}$ |
| **ARGOS** | **70%** $^{\mathbf{70.6}}_{\mathbf{69.8}}$ | **81%** $^{\mathbf{81.8}}_{\mathbf{80.0}}$ | **80%** $^{\mathbf{80.5}}_{\mathbf{78.8}}$ | **78%** $^{\mathbf{78.4}}_{\mathbf{77.7}}$ | **76%** $^{\mathbf{76.3}}_{\mathbf{75.8}}$ | **78%** $^{\mathbf{78.2}}_{\mathbf{77.7}}$ | **98%** $^{\mathbf{98.7}}_{\mathbf{97.9}}$ | 97% $^{98.2}_{96.3}$ | **97%** $^{\mathbf{98.1}}_{\mathbf{96.2}}$ |

| | CosmosQA | | | ESNLI | | | QUAIL | | |
|---|---|---|---|---|---|---|---|---|---|
| | M7B | L8B | L70B | M7B | L8B | L70B | M7B | L8B | L70B |
| SC20 | 84% $^{84.3}_{82.9}$ | 81% $^{81.3}_{79.7}$ | 90% $^{91.1}_{89.9}$ | **97%** $^{\mathbf{97.7}}_{\mathbf{97.1}}$ | **96%** $^{\mathbf{97.0}}_{\mathbf{96.3}}$ | **99%** $^{\mathbf{99.5}}_{\mathbf{99.2}}$ | 70% $^{71.0}_{67.9}$ | 68% $^{69.1}_{65.6}$ | 75% $^{75.6}_{72.4}$ |
| COT | 81% $^{81.4}_{80.1}$ | 76% $^{77.5}_{74.2}$ | 88% $^{88.7}_{86.9}$ | 96% $^{95.8}_{96.4}$ | 88% $^{86.9}_{88.4}$ | 99% $^{98.9}_{99.4}$ | 71% $^{71.9}_{68.8}$ | 65% $^{66.2}_{63.6}$ | 75% $^{75.6}_{72.4}$ |
| SAT-LM | 35% $^{37.5}_{34.8}$ | 35% $^{37.5}_{34.8}$ | 35% $^{37.5}_{34.8}$ | 49% $^{50.0}_{47.7}$ | 49% $^{50.0}_{47.7}$ | 49% $^{50.0}_{47.7}$ | 53% $^{55.0}_{51.6}$ | 53% $^{55.0}_{51.6}$ | 53% $^{55.0}_{51.6}$ |
| LoT-20 | 77% $^{77.2}_{76.8}$ | 75% $^{75.9}_{74.2}$ | 85% $^{85.7}_{84.3}$ | 71% $^{72.1}_{70.4}$ | 76% $^{77.5}_{75.7}$ | 75% $^{76.1}_{74.4}$ | 62% $^{63.6}_{59.9}$ | 56% $^{57.3}_{53.8}$ | 72% $^{73.1}_{69.9}$ |
| LLM-Tres | 73% $^{72.1}_{74.0}$ | 72% $^{72.7}_{70.9}$ | 71% $^{71.5}_{69.8}$ | 51% $^{52.5}_{50.8}$ | 51% $^{52.5}_{50.8}$ | 51% $^{52.5}_{50.8}$ | 63% $^{65.9}_{62.8}$ | 60% $^{62.1}_{58.8}$ | 58% $^{60.1}_{57.1}$ |
| ARGOS | 84% $^{84.3}_{82.7}$ | **83%** $^{\mathbf{84.0}}_{\mathbf{82.6}}$ | 90% $^{90.7}_{89.4}$ | 95% $^{95.6}_{94.9}$ | 96% $^{96.2}_{95.5}$ | 98% $^{98.0}_{97.4}$ | **82%** $^{\mathbf{83.4}}_{\mathbf{80.6}}$ | **82%** $^{\mathbf{83.4}}_{\mathbf{80.7}}$ | **80%** $^{\mathbf{81.8}}_{\mathbf{78.5}}$ |

**RQ1: How useful are the scoring and backbone-tracking elements?** In Table 2, we test the importance of two elements of ARGOS: (i) score thresholding and (ii) backbone computation. The ablation of each element in isolation results in a decrease in performance. In addition, the ablation of both results in a larger performance drop than even the sum of the two single ablations' decreases. The fully-ablated method, however, still shows strong performance relative to the next strongest

baseline (SC-20), highlighting the strength of the general concept behind the method. For further ablations, see Appendix E.

**RQ2: How often are ARGOS' added clauses useful or harmful?**  An important criterion when adding clauses is that they do not corrupt the logic of the problem, undesirably changing the outcome of the logic. It can be shown (see Appendix A) that so long as clauses are commonsensical, their addition will not corrupt the problem. However, it is possible that our method adds non-commonsensical clauses, since the commonsense scoring is not perfectly reliable. Given CLUTRR's strict structure, since we know the full knowledge base from which it was constructed, we can re-construct the full problems and test if ARGOS' added clauses corrupt the logical arithmetic such that a different answer is found for the logic problem. We find that on CLUTRR, ARGOS *never* corrupts a problem. It is then not surprising that ARGOS sees significant performance gains: added information should in principle never negatively affect a wholly rational reasoner's solution and so performance should only improve. It is also, of course, important that the added clauses contribute to the (correct) solution of the problem. In order to identify what information is important to the solution of the problem, we add the full relational reasoning rules to the SAT problem representing each CLUTRR example. We then extract the proof, taking all variables mentioned in the proof as important to solving the problem. We can then measure the number of problems for which at least one new variable is added, which is important to the proof. We find that ARGOS, on Llama 8B, identifies important new variables for 65% of the CLUTRR questions we test on.

**RQ3: Does ARGOS attribute more compute to harder problems? How does this affect the solution of harder problems?**  In Figure 5 (b), we examine the proportion of CLUTRR problems that are solved correctly by SC and ARGOS, over subsets of the dataset grouped by the number of ARGO iterations before termination. The error bars are 5/95% confidence intervals. As the number of ARGOS iterations increases, the problems become harder for SC to solve (indicated by a lower proportion of correct solutions by SC). This tells us that ARGOS' method of evaluating solvability is working as intended; harder problems are being assigned more computation. Another interpretation of this result is that problems which have more missing information, or for which the missing information is more difficult to infer, are attributed more ARGOS iterations (in order for ARGOS to find the necessary information). This is supported by the fact that the decrease in proportion seen in SC is not present for ARGOS: if SC's performance is dropping due to missing information, then ARGOS is successfully recovering the necessary missing information.

This ability to address the obstacles which cause SC performance to drop contribute to a large number of answers being flipped from incorrect (when solved by self-consistency) to correct (when solved by ARGOS). Changes to the answers caused by new information are more often than not in the right direction. On CLUTRR L70B, we find 112 correct and 35 incorrect flips. Figure 5 (a) shows the number of correct and incorrect flips ARGOS achieves. As the number of ARGOS iterations increases, both the correct and incorrect flip counts increase, but the correct flip counts increase much faster. For a closer look at confidence-score vs. iteration behavior, see Appendix K.

Figure 6 provides an example of a question from CLUTRR that is misclassified by self-consistency but flipped to correct by ARGOS. The COT seems confused, displaying its characteristic inability to plan out a proof: in steps 1-3 it provides disjoint pieces of information that neither follow from each other nor move towards the target conclusion. This confusion eventually leads to an incorrect step: "Shantel is Laura's aunt", resulting to an incorrect conclusion. ARGOS, after 3 iterations, provides several pieces of key information which would require at least one additional reasoning step to find, halving the necessary chain-length. For some examples in which ARGOS fails, see Appendix J. For results, evaluated by a human, on ARGOS and COT faithfulness on FOLIO, see Appendix C.

## 5.2 IMPACT OF IMPERFECT LOGICAL TRANSLATION

Here, we test if the assumption of perfect logical translation we make in our experimental procedure is justified via empirical result. In our experiments, we assumed that we started from a propositional logic formulation. Some datasets came with an official formulation, while for the others we translated from text using Claude Opus 4, filtering to remove failed translations. This was done to fairly evaluate the methods on abductive reasoning, regardless of the quality of translation. In general, logical translation is kept as a separate module to the proof planning/execution module in logical reasoning

Table 2: Ablations. We ablate elements of ARGOS: (i) the score thresholding, taking the first clause sampled at each iteration (ARGOS - No T), (ii) the backbone-tracking, generating prompts by randomly selecting two variables (ARGOS - No BB).

|  | FOLIO L8B |
| --- | --- |
| SC-20 | 71% $^{71.7}_{70.1}$ |
| ARGOS - No T | 79% $^{79.8}_{78.5}$ |
| ARGOS - No BB | 79% $^{79.4}_{78.4}$ |
| ARGOS - No Both | 76% $^{75.2\%}_{77.2\%}$ |
| ARGOS | **81%** $^{81.8}_{80.0}$ |

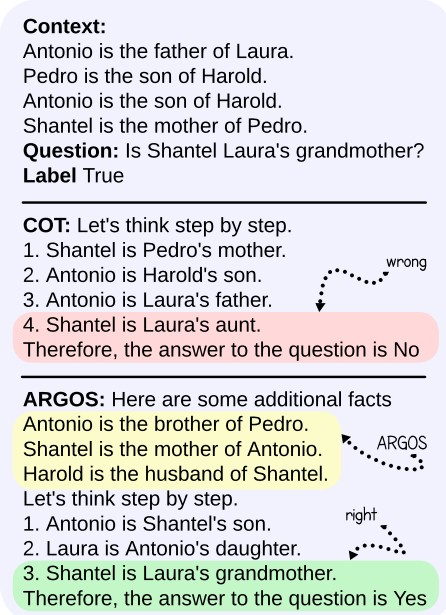

**Context:**
Antonio is the father of Laura.
Pedro is the son of Harold.
Antonio is the son of Harold.
Shantel is the mother of Pedro.
**Question:** Is Shantel Laura's grandmother?
**Label** True

**COT:** Let's think step by step.
1. Shantel is Pedro's mother.
2. Antonio is Harold's son.
3. Antonio is Laura's father.
4. Shantel is Laura's aunt.
Therefore, the answer to the question is No

**ARGOS:** Here are some additional facts
Antonio is the brother of Pedro.
Shantel is the mother of Antonio.
Harold is the husband of Shantel.
Let's think step by step.
1. Antonio is Shantel's son.
2. Laura is Antonio's daughter.
3. Shantel is Laura's grandmother.
Therefore, the answer to the question is Yes

Figure 6: COT vs ARGOS on a CLUTRR problem.

systems. Also, the translation task is intrinsically simpler for LLMs, since it is linguistic rather than cognitive. LLMs have already demonstrated strong abilities at logic translation (Yang et al., 2024), and are expected to continue improving faster than at reasoning. To validate this claim in our context, we re-tested ARGOS with Llama 8B on FOLIO using a translator, but including failed translations. Performance only decreased marginally, from 80% to 78%, still outperforming the next best method (SC at 71%). On QUAIL, ARGOS performance dropped from 82% to 73%, which while large relative to the drop on FOLIO still keeps ARGOS as the best performing method on QUAIL.

## 6 CONCLUSION

We have presented a method for addressing realistic natural-language logic problems, where "realistic" entails a need for abduction and commonsense. Whether neural or symbolic, we demonstrate empirically that existing methods struggle in this setting. The method we present addresses this weakness by *(a) balancing neural and symbolic elements and allowing them to speak to each-other*; and *(b) avoiding the commonplace design choice of heavily restricting the abduction-clause search space*. On both general and highly structured logic problems, our method demonstrates the power of a balanced neuro-symbolic approach, outperforming all existing work meaningfully.

**Limitations** A limitation of our work is that it is currently restricted to problems which are strictly True or False, eliminating cases where logic might be used to select an option from a list of choices, or cases where the correct answer is "Maybe". In our experimental work, we addressed the consequences this had on dataset selection by converting datasets to be True/False. The method could however be extended to multiple-choice questions by asking each question as an individual True/False question, combined with a decision heuristic for when no/multiple choices are determined True. Another limitation is that we restrict ARGOS to generating rules with up to two literals in the antecedent. While many-literal propositional formulas can often be decomposed into smaller ones, this may not always be the case and an ideal method would allow for large-antecedent generation. Thirdly, while our goal was to develop methods for open-source LLMs, the method would be more easily applicable if it did not require logit-level access. A potential future direction might be to convert the scoring system from a logit-based one to a verbalized score. Additionally, most benchmarks employed were modified to our setting, making the evaluated tasks at times artificial. Also, ARGOS sometimes depends upon self-consistency for problem solution. So, there are times when unfaithful or hallucinatory COTs will impact ARGOS' final prediction. Finally, this work focuses on forward chaining. A future direction may be backwards-chaining approaches to abductive reasoning.

## REPRODUCIBILITY STATEMENT

In the supplementary material, we provide our full code which was used to implement and benchmark our method as well as the baselines. The code also includes data processing steps. We took great care to include in the Appendix, as well, detailed descriptions of our algorithm and our prompts. While our human modification of FOLIO text is not provided, the process for generating it is described carefully in the Appendix.

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

## A   ABDUCTIVE LOGIC PROBLEMS ARE WELL-DEFINED

In this section we prove that the solution of an abductive propositional logic problem, given in Section 3, does not depend on the choice of commonsense set $\mathcal{C}$.

**Proposition 1.** *Let $\mathcal{P}$ be a set of premises, $Q$ a query proposition, and $\mathcal{C}_1, \mathcal{C}_2$ subsets from commonsense set $\mathcal{C}$ of additional propositions such that $(\mathcal{P} \wedge \mathcal{C}_1) \vdash L_1$ and $(\mathcal{P} \wedge \mathcal{C}_2) \vdash L_2$ for literals $L_1, L_2 \in \{Q, \neg Q\}$. If $\mathcal{P}$ is consistent with $\mathcal{C}$ ($\mathcal{P} \wedge \mathcal{C} \nvdash \bot$) then $L_1 \leftrightarrow L_2$.*

*Proof.* Let's say $L_1 \not\leftrightarrow L_2$: without loss of generality we can take $(\mathcal{P} \wedge \mathcal{C}_1) \vdash Q$ and $(\mathcal{P} \wedge \mathcal{C}_2) \vdash \neg Q$. So, $(\mathcal{P} \wedge C_1 \wedge C_2) \vdash (Q \wedge \neg Q) \vdash \bot$. But $C_1, C_2 \subset C$, so therefore $(\mathcal{P} \wedge C) \leftrightarrow (\mathcal{P} \wedge C_1 \wedge C_2 \wedge [C \setminus C_1 \cup C_2])$, so $\mathcal{P} \wedge C \leftrightarrow \bot \wedge [C \setminus (C_1 \cup C_2)]$, so $\mathcal{P} \wedge C \vdash \bot$. This contradicts our assumption that $\mathcal{P} \wedge \mathcal{C} \nvdash \bot$. $\square$

## B   COST DISCUSSION

While in theory COT generation is meant to be done until an answer is found, in practice it is necessary that an upper-limit on number of tokens generated is enforced. This is in case (a) the LLM continues generating past its answer, or (b) the LLM goes off-track and never answers the question. In any case, this means that each COT generation will be, at worst-case-assumption, equal in cost. In addition, the various method-specific LLM generations that are employed require small token-limits relative to COT, and so the number of COT calls made dominates the total number of tokens generated by any method. Additionally, as problems get harder and more logically complex, necessary COT generation length increases, making this even more true. So, we can say that cost for each method scales in proportion to the number of COTs generated. For example, Self-Consistency takes two hours longer to run than AROGS on FOLIO with Llama 8B, despite requiring more total LLM calls (where we include scoring and literal-generaiton calls in our count). This is because the average number of COT-specific calls is lower for ARGOS than SC, and the scoring and literal generation calls are much shorter than COT calls. In our implementation, we generate at most 25 tokens for literal-generation, 1 token maximum for scoring, and 300 tokens maximum for COT generation. Given this, budgeting method costs in terms of COT calls is well-justified. For SC and COT, the cost evaluation is trivial: COT always makes 1 COT call and SC makes a fixed number of COT calls, specified as a hyper-parameter. Similarly, LOT makes some small generative calls followed by SC, so its number of COT calls is fixable. LLM-Tres makes no COT calls, and neither does SAT-LM. ARGOS' cost varies according to the entry, but its hyper-parameters (number of COT calls per-iteration and threshold/annealing constants) can be set such that its average number (or worst-case) number of calls is less than a budget. A summary of method cost in terms of COT calls is provided in Table 3. In Figure 7, we show a histogram of the individual problem costs incurred by ARGOS with Llama 8B, expressed in terms of the number of COTs required per problem. For most problems, only 10 COTs are required. This allows ARGOS to exceed the on-average cap for problems requiring more ablation or deeper search. If we compose a new dataset of only the hardest problems, for example the CLUTRR problems for which ARGOS takes 8-10 ARGOS steps (the final bar in Figure 5(b)). Testing ARGOS, limited at 20 COTs per-problem (strictly, not on average), we see that its performance drops from 65% to 54% on these problems, whereas SC performs at 40$ with these problems. This indicates a clear and steep cost-performance tradeoff for ARGOS, but even with strict cost limitations ARGOS outperforms SC.

Table 3: Average number of COT calls required by each method.

|  | Cost (Avg # COT) |
| --- | --- |
| COT | 1 |
| SC | 20 |
| LOT | 20 |
| SAT-LM | 0 |
| LLM-Tres | 0 |
| ARGOS | 18.4 |

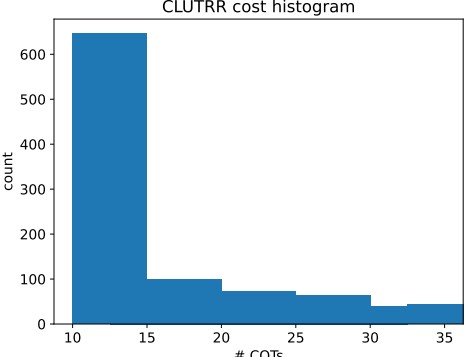

Figure 7: Histogram of individual problem costs to ARGOS-Llama 8B on CLUTRR

## C   MANUAL HUMAN FAITHFULNESS EVALUATION

For COT, we manually checked the chains which result in the correct final answer, generated with Llama 8B. For ARGOS, if ARGOS made the correct final prediction using the SC output, we will check one of the chains from the final SC call. If ARGOS made the final prediction using the symbolic solver, the reasoning is considered correct if none of the added clauses are incorrect (the reasoning itself must be beyond reproach as it is executed by a solver). Additionally, for ARGOS, we evaluate the usefulness of the added information in solving the problem, and whether the added information is new to the problem. We find that Llama 8B generates faithful COT reasoning processes when it gets the answer right 72% of the time. We find that ARGOS-L8B generates a faithful reasoning process when it gets the answer right 85% of the time, showing that in general ARGOS-L8B is more faithful than pure-COT based methods (i.e. COT, SC). Three potential explanations for this result, in order from least to most in terms of their strength in justifying our method, are that (1) The logical content of ARGOS's augmented prompt, regardless of content, incites a more logical structure in the LLM generation. (2) ARGOS successfully extracts the key elements of the problem, stabilizing the LLM and making it less likely to become unfaithful or to hallucinate. (3) ARGOS adds new information which allows us to solve otherwise difficult problems. Empirically, we find that ARGOS-L8B adds at least one piece of information necessary for the generating the faithful proof 85% of the time. This seems to lend credibility to explanation (2). Additionally, ARGOS adds new information to the text-problem 72% of the time, which lends some credibility to explanation (3), which would strongly justify the extend of our method, whose goal is explicitly stated as searching for new and crucial commonsense. Most interestingly, we find that 100% of the time in which we solve symbolically, the information added by ARGOS is faithful, useful and novel. This is perhaps not surprising, since to achieve the contrary, we would have to either add exactly contradictory information to the true proof. Nonetheless, it is an extremely satisfying finding, as it shows that when we are able to avoid typical symbolic robustness issues such as symbol mismatch, ARGOS maintains the rigour characteristic of symbolic methods.

# D    ALGORITHM DESCRIPTION

In this section we provide a detailed description of our ARGOS algorithm. The main procedure is summarized as Algorithm 1, which uses the `find_new_commonsense` subroutine in Algorithm 2.

---

**Algorithm 1** ARGOS

---

**Require:** premises $\mathcal{P}$, query $Q$, SC sample-count $k$, scoring threshold $\tau \in (0, 1]$,
self-consistency threshold $\gamma \in (0, 1]$ and decay $\alpha \in (0, 1]$
1: commonsense set $\mathcal{C} \leftarrow \{\}$
2: **while** $\gamma > 0$ **do**
3:      *// Attempt solving with the SAT solver*
4:      sat_conclusion, backbone $\mathcal{B} \leftarrow$ `sat_solve`$(\mathcal{P} \wedge \mathcal{C} \vdash Q, \neg Q)$
5:      **if** sat_conclusion is $(\mathcal{P} \wedge \mathcal{C}) \vdash Q$ or $(\mathcal{P} \wedge \mathcal{C}) \vdash \neg Q$ **then**
6:          **return** $\mathcal{C}$, sat_conclusion
7:      **end if**
8:      *// Else attempt solving with the LLM ($k$-shot self-consistency)*
9:      llm_conclusion, llm_confidence $\leftarrow$ `llm_solve`$(\mathcal{P} \wedge \mathcal{C} \vdash Q, \neg Q)$
10:     **if** llm_confidence $> \gamma$ **then**
11:         **return** $\mathcal{C}$, llm_conclusion
12:     **end if**
13:     *// Else find a new commonsense proposition to add to the pool*
14:     $C \leftarrow$ `find_new_commonsense`$(\mathcal{P}, \mathcal{C}, \mathcal{B}, \tau)$
15:     **if** $C$ is not None **then**
16:         *// New commonsense has been found, we try again with an enlarged $\mathcal{C}$ and smaller $\gamma$*
17:         $\mathcal{C} \leftarrow \mathcal{C} \wedge \{C\}$
18:         $\gamma \leftarrow \gamma - \alpha$
19:     **else**
20:         *// We failed, return best guess*
21:         **return** $\mathcal{C}$, llm_conclusion
22:     **end if**
23: **end while**
24: *// We ran out of time, return best guess*
25: **return** $\mathcal{C}$, llm_conclusion

---

**Algorithm 2** `find_new_commonsense`

---

**Require:** premises $\mathcal{P}$, commonsense $\mathcal{C}$, backbone $\mathcal{B} = \text{backbone}(\mathcal{P} \wedge \mathcal{C})$, scoring threshold $\tau \in (0, 1]$
1: **for** $L_1 \in \mathcal{B}$ from highest to lowest $\text{score}_{\mathcal{B}}(L_1)$ **do**
2:      **for** $L_2 \in \mathcal{B}$ from highest to lowest $\text{score}_{\mathcal{B}}(L_2)$ **do**
3:          **for** $L_{\text{right}}$ **in** `llm_generate`$_{\mathcal{P} \wedge \mathcal{C}}(L_1 \wedge L_2 \to ?)$ **do**
4:              commonsense_score $\leftarrow$ `llm_commonsense_score`$(L_1 \wedge L_2 \to L_{\text{right}})$
5:              relevance_score $\leftarrow$ `llm_relevance_score`$_{\mathcal{P} \wedge \mathcal{C}}(L_1 \wedge L_2 \to L_{\text{right}})$
6:              **if** commonsense_score $> \tau$ **and** relevance_score $> \tau$ **then**
7:                  *// We found a new relevant commonsense clause*
8:                  **return** $L_1 \wedge L_2 \to L_{\text{right}}$
9:              **end if**
10:         **end for**
11:     **end for**
12: **end for**
13: *// We failed to find a new relevant commonsense clause*
14: **return** None

---

# E ABLATIONS

|  | FOLIO 8B | CLUTRR 8B |
|---|---|---|
| ARGOS-Symbolic | 59% | 72% |
| ARGOS | 81% | 76% |
| SC | 71% | 69% |
| LLM-Tres | 63% | 51% |

Table 4: Ablating the SC-solver on ARGOS. ARGOS-Symbolic denotes the ablated version of ARGOS.

In Table 4, we ablate the SC-solver, solving problems with only a symbolic solver. We impose a threshold of 100 proposed rules, since the previous bounding system was in terms of SC confidence. We find that removing the SC option only somewhat hurts ARGOS performance on CLUTRR, but has a catastrophic effect on FOLIO. This is not at all surprising, since CLUTRR is far simpler and more logically structured than FOLIO. Interestingly, we find that LLM-Tres performs slightly better than ARGOS-Symbolic on FOLIO. On investigation, we found that propositions with a single-literal antecedent were generally enough to solve FOLIO problems, and so LLM-Tres' methodological restriction to such rules benefits it, whereas in CLUTRR where nearly all necessary propositions have two literals in the antecedent, LLM-Tres is reduced to guessing. This massive performance gap which we see on ARGOS in the more linguistic and more logical datasets illustrates the need for balanced, neuro-symbolic approaches.

|  | FOLIO 8B |
|---|---|
| No Commonsense | 79% |
| No Contextual | 80% |
| Full AROGS | 81% |

Table 5: Ablating individual score thresholds

In Table 5, we ablate separately the commonsense and contextualness scoring components. We find that both are necessary. This makes sense, since while they may have some overlap in terms of the rules which they serve to reject (for example, gibberish output would be neither commonsensical nor contextually relevant), it is easily conceivable that the sets of proposed propositions which they reject correctly is not totally overlapping.

# F ALGORITHM LLM DETAILS

In this section we provide details about the parts of the algorithm that involve interactions with the large language model.

## F.1 SOLVING WITH 5-ROUND SELF-CONSISTENCY (`LLM_SOLVE`)

This subroutine aims to establish whether the premises and commonsense entails the query, i.e. $(\mathcal{P} \wedge \mathcal{C}) \vdash Q$ or $(\mathcal{P} \wedge \mathcal{C}) \vdash \neg Q$. This routine always returns an answer, but can make mistakes. We few-shot prompt the LLM 5 times with the prompt in Table 6.

Table 6: COT prompt

```
Here are some facts and rules: [premises 𝒫]
Here is some additional info we found: [commonsense 𝒞]
True or false: [query 𝑄]?
Answer:
```

Each call returns an answer $a_1, \ldots, a_5 \in \{\text{True}, \text{False}\}$, with a certain confidence $c_1, \ldots, c_5 \in (0, 1]$. The algorithm returns the most common answer (True or False), and a total confidence score given by

the sum of confidences of the most common answer $a^*$, divided by 5:

$$c^* = \frac{1}{5} \sum_{i=1}^{5} c_i \mathbb{1}[a_i = a^*].$$

## F.2 GENERATING NEW COMMONSENSE LITERALS $L_{\text{RIGHT}}$ (`LLM_GENERATE`)

This subroutine aims to find a plausible right-hand-side literal $L_{\text{right}}$ for a proposition $L_1 \wedge L_2 \to L_{\text{right}}$. The new literal might potentially involve new variables not previously seen in the problem. We use a slightly different prompt for CLUTRR and for the others, because of CLUTRR's more distinct structure. Anecdotally, we found that CLUTRR's more consistent linguistic structure allowed for prompt formats which were very straightforward. For more linguistically complex datasets, however, more robust prompt formatting was required. For example, asking if a rule seems contradictory was more robust in situations where a rule was ambiguous without context (i.e. student(Rina) implies like_coffee(Rina), vs. Mom(Hannah, Sam) and Sibling(Sam,Mary) implies Mom(Hanna, Mary)).

Table 7: Clause generation prompt for all datasets but CLUTRR. $e$ is an entity appearing in $L_1$ or $L_2$.

```
Fill in the blank with a known predicate: [L₁ ∧ L₂] implies
____([e]).
Known predicates are: [all predicates appearing in the premises
𝒫 and the commonsense 𝒞]
Answer:
```

Table 8: Clause generation prompt for CLUTRR. $e_1$ and $e_2$ are entities appearing in $L_1$ or $L_2$.

```
If [L₁ ∧ L₂] then ____([e₁],[e₂]). Fill in the blank.
Answer:
```

Thus, for example, in FOLIO if $L_1 = drinksCoffee(Rina)$ and $L_2 = Loves(Mary, Sam)$, then we would make three calls, one for $drinksCoffee(Rina) \wedge Loves(Mary, Sam) \to F(Rina)$, $drinksCoffee(Rina) \wedge Loves(Mary, Sam) \to F(Mary)$ and $drinksCoffee(Rina) \wedge Loves(Mary, Sam) \to F(Sam)$ respectively. With such calls, the method might return the set $\{productive(Rina), hasFeelings(Mary), isLoved(Sam)\}$, for example.

## F.3 SCORING PROPOSITIONS $L_1 \wedge L_2 \to L_{\text{RIGHT}}$ FOR COMMONSENSE (`LLM_COMMONSENSE_SCORE`)

This procedure uses the LLM to score how much our new proposition $L_1 \wedge L_2 \to L_{\text{right}}$ is likely to be commonsense. In detail, we ask the LLM whether, without any context, the rule seems contradictory (FOLIO/ProntoQA) or true (CLUTRR). We record the logits of the "Yes" and "No" tokens following the prompt, and we return as commonsense score $P[\text{No}] = \exp(\text{logit}_{\text{No}})/(\exp(\text{logit}_{\text{Yes}}) + \exp(\text{logit}_{\text{No}}))$, except for CLUTRR where we return $P[\text{Yes}] = \exp(\text{logit}_{\text{Yes}})/(\exp(\text{logit}_{\text{Yes}}) + \exp(\text{logit}_{\text{No}}))$ since the question is inverted.

Table 9: Commonsense scoring prompt for FOLIO/ProntoQA.

```
Does the following rule seem contradictory?
Rule: [L₁ ∧ L₂ → L_right]
Answer:
```

## F.4 SCORING PROPOSITIONS $L_1 \wedge L_2 \to L_{\text{RIGHT}}$ FOR CONTEXT-RELEVANCE (`LLM_RELEVANCE_SCORE`)

This procedure uses the LLM to score how much our new proposition $L_1 \wedge L_2 \to L_{\text{right}}$ is likely to be relevant to the context. This helps eliminate propositions, like "The sky is blue", that are

Table 10: Commonsense scoring prompt for CLUTRR.

```
Does the following rule seem true?
Rule: [L₁ ∧ L₂ → L_right]
Answer:
```

true and commonsense but unlikely to help prove our query. In this case, we use the same prompt for all datasets. We record the logits of the tokens "Yes" and "No" following the text, and return $P[\text{Yes}] = \exp(\text{logit}_{\text{Yes}})/(\exp(\text{logit}_{\text{Yes}}) + \exp(\text{logit}_{\text{No}}))$ as relevance score.

Table 11: Context scoring prompt for all datasets.

```
Here are some facts and rules: [premises 𝒫 and commonsense 𝒞]
Does the following new rule seem contextually relevant to the
facts and rules? [L₁ ∧ L₂ → L_right]
Answer:
```

### F.5 MANUAL EVALUATION OF SCORING MODULE

We have conducted a human review of 50 ARGOS-proposed rules and their corresponding scores, for Llama 8B on FOLIO. We found that, by classifying with the decision rule of thresholding at 0.3 for commonsense-ness and contextual-ness separately, the commonsense binary classification is correct 76% of the time, and the contextualness classification accuracy is correct 91% of the time. We find that the scores are not in general well-calibrated, but this is not needed for the thresholding that we do.

## G IN-DEPTH LITERATURE REVIEW

Since 2022, when Wei et al. (2022) found that models had the capacity to solve (at the time) difficult reasoning problems by simply prompting the model to output a detailed rationale (a "chain of thought" (COT)) before making a decision, there has been significant interest in leveraging/improving LLMs' capacity for reasoning.

**Semantic Logic Parsers** More recently, a series of methods were proposed which effectively bypassed the reasoning process by prompting LLMs to translate the given input to symbolic language (parsing the logic) and then using external, programmatic solvers to solve the problem in the symbolic space. F-COT, proposed by Lyu et al. (2023) and SAT-LM, proposed by Ye et al. (2023) are two contemporaneous works which prompt the LLM to translate the given problem into its corresponding symbolic language, to be solved by the appropriate solver. Logic-LM, proposed by Pan et al. (2023), also published around the same time, includes a self-refinement step in order to catch mistranslations and to re-translate them, but this module provided only minor improvements.

The logic-parsing strategy proved extremely effective, converting reasoning tasks into the more linguistic translation task. Since the algorithmic tools never make mistakes, if the translation is accurate, then the solution will be too. These methods, however, rely upon the strong assumption which goes widely unacknowledged that all necessary information is provided at input. By motivation, this assumption implies a well-informed, precise, and careful end-user. This critical assumption greatly injures the applicability and generalizability of these symbolic methods. In order to address this, ARGOS aims specifically to provide missing information by exploring the logical space and leveraging logical tools. Xu et al. (2024), argue that using external solvers is not relevant to the LLM's actual reasoning capacity and present a method which still parses the text into symbolic language, but uses the LLM as the symbolic solver by inputting the symbolic expressions directly to the LLM. In fact, this was an alternative presented in SAT-LM, and was shown to be weaker than directly using an external solver in cases where the problems are fully defined (which is the case in the setting chosen by both works).

**Deductive Logical Algorithms**   Given the findings discussed above that LLMs were efficient logic-parsers, the door was opened to more agentic algorithms which operate in the logic-space (as opposed to the textual space). Algorithms were proposed by Kazemi et al. (2023) and Lee and Hwang (2024) which attempted to reason backwards through the problem in logic space, using the LLM after each deduction step to choose the next deduction. These methods proved to perform worse than the previous parsing methods as they relied on the same assumptions of completeness in the input, but also required the LLM to greedily trace a reasoning path, opening another avenue for errors. This step is unnecessary since logic tools are able to search exhaustively over the solution space at very low cost.

**Clause-generative Algorithms (Abduction)**   In response to the literature's inability to address cases where the completeness assumption fails, some work was proposed which explicitly aims to generate logical clauses in the textual space. The very similar algorithms proposed by Liu et al. (2024) and Wang et al. (2022) first translate the text to logic and then produce some new clauses which are deducible via classical logic given the logical translation of the input. By then translating the newly deduced logic back to text, the textual representation of the logic is now richer from a linguistic point of view (although no new information was added to the underlying logic), and then COT is used to solve the augmented problem in text-space. In the logical sense, this method is not truly abductive as no new logical information is produced.

Instead of producing clauses via deduction on the input, Toroghi et al. (2024) proposed a method which leverages LLMs' knowledge of commonsense to supply missing clauses during the reasoning process. Given the logical translation of the input text, the space of all 2-variable clauses which are possible to construct using the variables given in the question is explored exhaustively, with each rule being given a probability of being true by the LLM. Reasoning paths are formed with the various rules.

This method suffers from a rigidity regarding the search space: due to the exhaustive search the space must be restricted to only 2-variable rules constructible from the input. Often the missing information from logic problems may include variables which are not named in the problem and so are unseen in the abduction input. In other cases, the necessary rules might be of different sizes. These cases are easily seen even in the commonly used datasets for logic-reasoning evaluation. For example, CLUTRR Sinha et al. (2019), which is a dataset of reasoning problems related to family relationships, requires 3-variable rules of the form $mom(A, B) \land sister(B, C) \rightarrow mom(A, C)$. This algorithm is by construction incapable of producing such information. The evaluation procedures presented by Toroghi et al. (2024) use simple and rigidly structured data such that only simple 2-variable rules constructed from existing variables are required.

Since the algorithm output is dependent on the logical proof, if the necessary clauses cannot be found then a full proof will never be found, and the algorithm is forced to provide either no output or a guess. In contrast to these works, ARGOS aims to introduce logical clauses which provide new information about potentially unseen variables and which can be of varying sizes. In order to accommodate the very large search space, we introduce several innovative methods to dynamically select sub-spaces and to efficiently search them. Our proposed method aims to build its solution from both the language-reasoning space and the logic space in order to leverage the exactness of logical tools while remaining robust to failures to find the necessary logical clauses.

**Generalist Reasoners**   Despite much research, Chain-of-Thought (COT), proposed by Wei et al. (2022), remains one of the most generally applicable and robust reasoning approaches Plaat et al. (2024). Thus, much research has been done on how to augment the COT process (the previously discussed work by Liu et al. (2024) could be viewed as part of this category). While the focus of our paper is logical algorithms, there are some generalist methods which cannot be ignored. A simple but highly effective approach known as self-consistency (SC), proposed by Wang et al. (2023), prompts via COT several times and takes the mode of the outputs as the prediction. This method benefits from adding further robustness by smoothing potential outlier errors which might be present in a few chains, and is easily scalable according to computational budget by choosing the appropriate number of samples to take. However, the marginal return decreases as the number of sampled chains increases, as is shown in the original paper Wang et al. (2023). COT and SC's poor performance in proof planning (Saparov and He, 2023) motivated Tree of Thoughts (TOT), proposed by Yao et al. (2023), in which a reasoning tree is hand-crafted for a problem and then the LLM is prompted to

explore this tree for likely paths. This method demands a very specific tree topology and exploration designs for each task and so is more of a general framework than an explicit method applicable to logical reasoning. The tree structure of logical reasoning problems even in the same dataset are highly varied. Given this, TOT is poorly suited to logical reasoning settings. There are many works which go beyond TOT in terms of exploring potential chains. Most recently, Xue et al. (2024) proposed a recursive method which also conducts a tree-like search, but allows for dynamically-structured trees.

## H   USED DATASETS

The following is a description of the datasets which we have used in our experiments.

**ProntoQA (Saparov and He, 2023)**   ProntoQA is a dataset which comes in three types of groundings over the same symbolic structure, which is a string of single-variable modus ponens operations. One of these types is a hand-crafted grounding designed to be true according to commonsense. This dataset is, at this point, fairly easy for language models to handle. However, it remains a common dataset in logical reasoning, and comes with the convenience that a simple random removal of inference rules included in the problem will build abduction cases, as all rules are commonsense. There are 59 problems in the dataset.

**CLUTRR (Sinha et al., 2019)**   CLUTRR is a dataset of family-relational reasoning problems in which some family relations (i.e. "Sam is the mother of John") are given in the form of simple stories (i.e. "John went with his mom Sam to the mall"). Traditionally, the task in CLUTRR is to deduce the relationship between two people, given the context. In order to structure the problem as a true/false classical logic problem, however, we restructure the task to determine if a given relationship between two people is true or not. Practically, we construct the labels by taking the ground-truth relationship as the query 50% of the time (so the new label is "True"), and taking a random other relationship between the given two people ("False") the other 50% of the time, in order to balance the dataset. While the task is naturally abductive in that the input contexts do not include abstract or even grounded relational inference rules (i.e. "if A is the mother of B then B is the child of A"), most symbolic methods rely upon a practitioner to hand-craft a knowledge base of relational rules which are appended to each problem in the dataset. Simply by forbidding this provision, the task becomes truly abductive for the reasoning model. There are 1000 problems in the dataset.

**FOLIO (Han et al., 2024)**   FOLIO is a dataset of logic problems which were hand-crafted by "expert annotators". The dataset is the most diverse of the three we use, both linguistically and structurally. While not perfectly commonsensical, it is generally based in commonsense simply because the annotators were humans who exist in and tend to operate within real-world contexts. Given this pseudo-commonsensicality, random removal of rules to introduce commonsense abduction is not an option, as non-commonsense may be removed from the context. Thus, we engaged human annotators to replace randomly selected phrases from problems with semantically equivalent expressions, so that each replacement would require a minimum of one new rule, indicating that the replaced phrase is implied by its replacement. For example, "NBA Player" → "Pro Basketball Player". Annotators were instructed to reject problems where no replacement could easily be found, and some annotations failed to impact the problem due to either weak replacements or non-interaction with the solve-path of the problem. These problems were discarded, leaving us an abduction-variant of FOLIO with 108 True-or-False problems.

**ESNLI (Camburu et al., 2018)**   ESNLI is a dataset of premise-conclusion pairs, stemming from the human-explanation NLI field. A machine is asked to explain how the premise yields or contradicts the conclusion. We adopted this dataset by inverting the task, so that given the premise and conclusion the machine must determine whether the conclusion is entailed (True) or contradicted (False). The task naturally ensures that abduction is necessary, as we leave out the human explanation with which the pairs are annotated. The dataset, on inspection, is mostly common-sensical.

**CosmosQA (Huang et al., 2019)**   CosmosQA is an MCQA dataset designed to test machine reading comprehension. We adapt it to our setting by constructing problems for which the answer is "False" by taking questions for which "None of the Above" is the correct answer and randomly selecting

another of the answer choices to be the query (for problems for which the answer is "True", the process is obvious). The dataset is mostly commonsensical.

**QUAIL (Rogers et al., 2020)**  QUAIL is an MCQA dataset also designed to test machine reading comprehension. It is derived primarily by scraping wiki and forum pages on online, and so it often contains artifacts such as timestamps or CSS formatting quirks. We adopt it from MCQA to True/False in an identical fashion to CosmosQA. Becasue of its provenance, questions are often extremely vague; the larger context of the webpage from which the problem comes is not included but is often key to answering the problem. On manual solution of QUAIL problems, the intuitive approach is often to start by inferring the original context of the scraping, making the dataset of high value for abductive settings and for robust evaluation of commonsense flexibility.

## I  UNUSED DATASETS

While there are many logical-reasoning-related datasets available, many are unsuitable because they are either not truly logically-structured or are not commonsensical. Here, we will list some of the most commonly used datasets for logic-adjacent applications and explain their weaknesses/unsuitability for our setting.

**LogiQA**  While LogiQA (Liu et al., 2021) is generally commonsensical and logically themed, its questions do not in fact impose an immediate logical problem. In fact, many of the questions are in fact *meta-logical*, in that they ask questions about the underlying logic of the text. For example: "Which of the following makes the same logical mistake as above". These questions could indeed have a formal-logical re-framing, but this would require far more logical aptitude than is currently held by language models, and hand-translating LogiQA questions to logic problems is too time-consumptive.

**RECLOR**  RECLOR (Yu et al., 2020) suffers the same weakness as LogiQA. Questions are meta-logical or ask for subjective qualifications regarding some described commonsense logic. Again, this type of question both fails to evaluate true logical reasoning in real-world contexts, and proves problematic for careful evaluation given the inconsistency of the task in that different questions ask different things of the reasoner.

**Soft Reasoner**  The Soft Reasoner dataset (Clark et al., 2021) is strictly logical, but is plainly non-commonsensical by construction. The logical problems are constructed without consideration of commonsense or real-world contexts. The dataset is constructed by building clauses from variables/predicates which are randomly selected from a hand-selected bag of words.

**LogicNLI**  LogicNLI (Tian et al., 2021) suffer from the Soft Reasoner dataset's weakness to an even greater degree - while the problems are also randomly generated and do not comply with commonsense, they also often do not comply with grammar. For example, phrases such as "Quinlan does not entire" appear frequently. While this may suit the authors' aims of producing arbitrary text as a stand-in for symbolic logic, it is not amenable to the evaluation of real-world logical reasoning in human contexts.

**ProofWriter**  ProofWriter (Tafjord et al., 2021) again suffers from the same weakness which all semi-random auto-generated datasets suffer: non-commonsenseness. By picking clauses randomly by sampling bags of predicates, no guarantees can be made on the realism of the data. Examination of the dataset will show that the "facts" described in problem contexts range from unlikely to non-sensical - the very first problem includes a rule "All red things are rough". Within the real world this is obviously not true, as we can find examples of red things which are not rough. Of course, it was not the dataset creators' aim to build a commonsensical dataset and so this is of little surprise.

## J  CONFUSING PROBLEMS FOR ARGOS

Here' we give some examples of problems which confused ARGOS. Note that all of the problems provided are examples in which the given premises are at least somewhat contradictory with commonsense, breaking the setting.

Table 12: Confusing since Fido is typically specifically a dog's name (and not a cat's). If we ask the LLM if Fido is a dog, with no context, it will say yes.

```
All tigers are cats. No cats are dogs. All Bengal tigers are
tigers. All huskies are dogs. Fido is either a Bengal tiger or a
cat.
True or False: Fido is a husky animal.
```

Table 13: Confusing since Detroit City is probably not a horse according to commonsense.

```
Detroit City is a horse. Some horses are racehorses. If a horse
falls in a race, it poses risks to its rider. Detroit City fell
in a race. A horse is a racehorse if it is in a race.
True or False: Detroit City has been in multiple races.
```

Table 14: Confusing because (1) edible can refer to beverages when the contextual distinction is between safe and unsafe for consumption, but not when the distinction is between eating and drinking; (2) Coke is not apple juice.

```
All drinks on the counter are edible. All juices on the counter
are drinks. Orange juice is a type of juice. Everything on the
counter is either orange juice or apple juice. All apple juices
on the counter are sweet. The coke is on the counter and if the
coke is apple juice, then the coke is a drink. If the coke is
not apple juice, then the coke is not edible.
True or False: The coke is edible and sweet.
```

## K    SOLVABILITY PROGRESSION ILLUSTRATION

In this section we provide figures for CLUTRR, CosmosQA and QUAIL, illustrating how ARGOS' SC-solvability measure progresses per-question, over a 100-question clipping from each dataset. Positive % values indicate a positive classification, and negative the like.

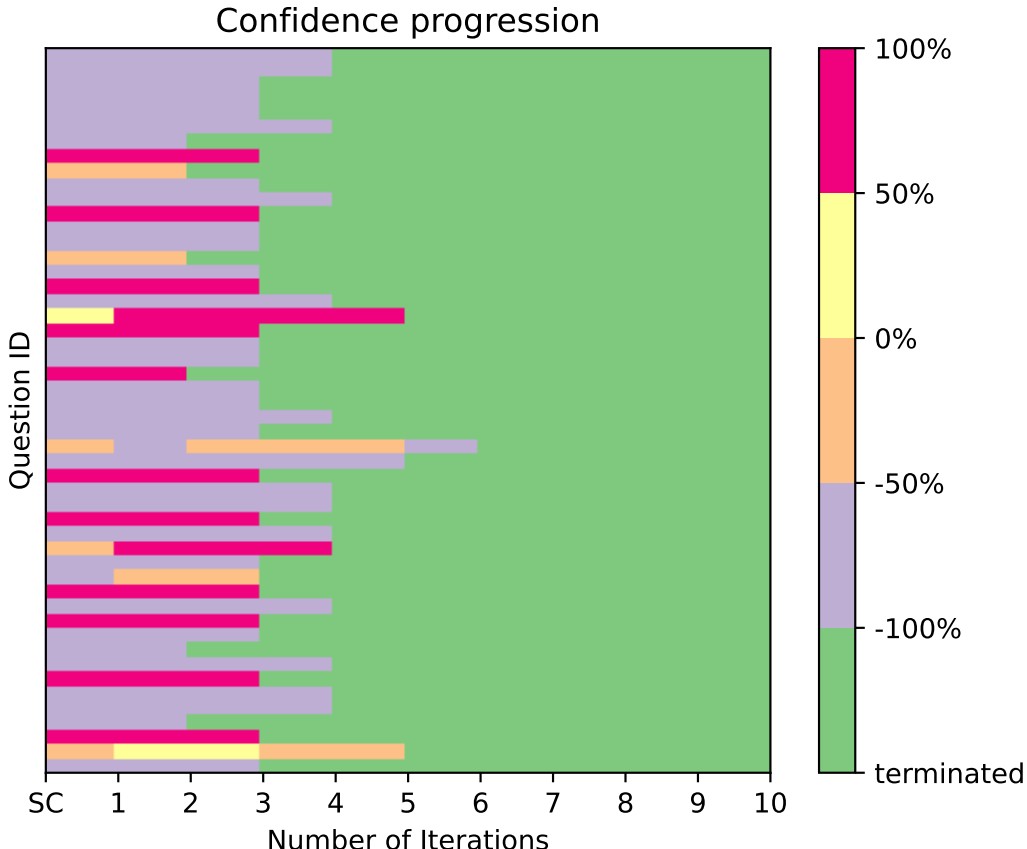

Figure 8: CosmosQA: This dataset, being mostly solved by vanilla SC, sees little fluctuation and exits from ARGOS early. With that said, we still see a flip from low-confidence negative to high-confidence positive.

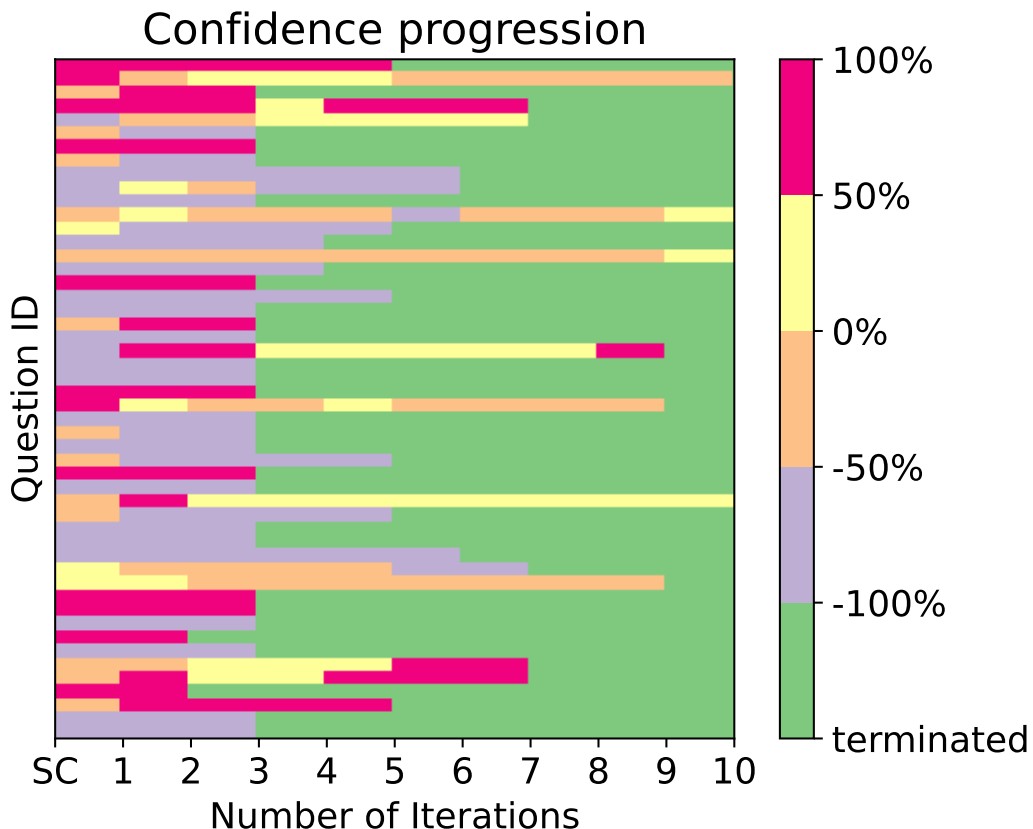

Figure 9: CLUTRR: We see a significant amount of confidence fluctuation and flipping, indicating that meaningful elements within the logic of the problem are being modified by ARGOS in order to affect the answer. This is not surprising, since our construction of generated propositions as taking purely backbone variables as antecedents ensures that added ARGOS propositions will be effectual.

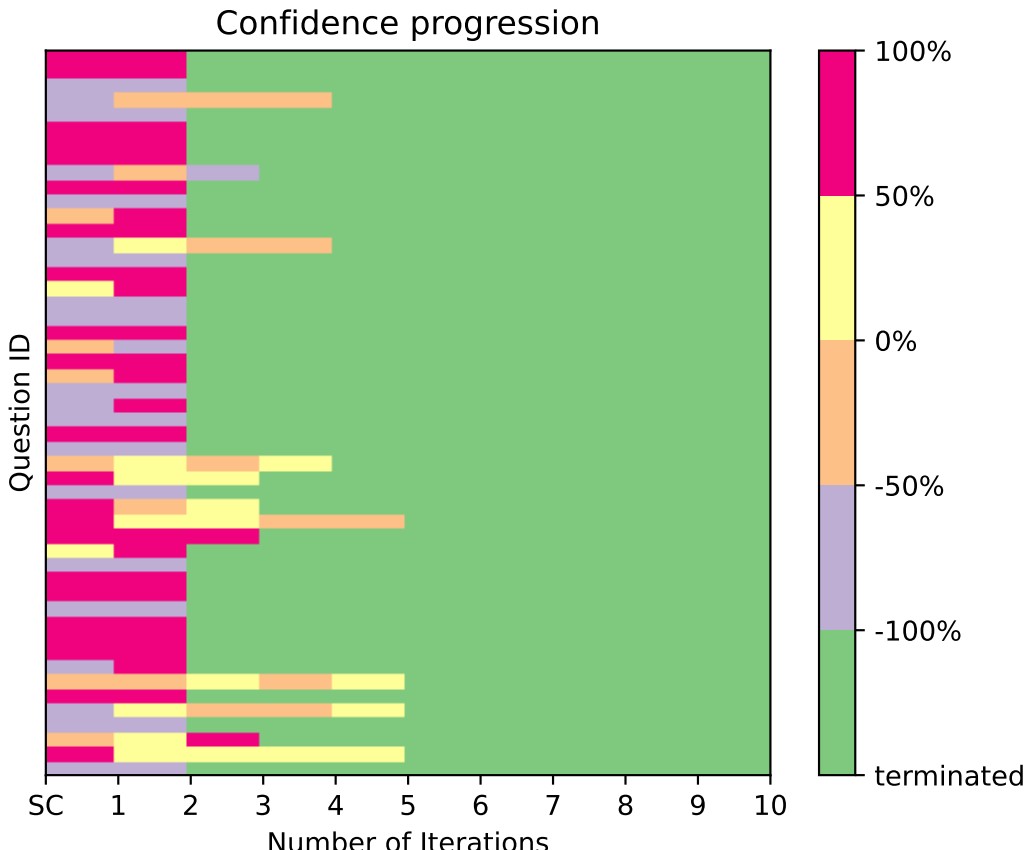

Figure 10: QUAIL: despite ARGOS's strong performance on QUAIL relative to SC, we find that in fact few ARGOS iterations are necessary. While QUAIL is made complicated for language models by its irregular form and often disjoint nature, its logical structure (while very ambiguous) is simple, meaning that we can solve QUAIL problems with only a small number of well-chosen propositions.

