# OpenReview forum: "A Balanced Neuro-Symbolic Approach for Commonsense Abductive Logic"
_ICLR.cc/2026/Conference — ICLR 2026 Poster_

### Official Review · Reviewer_L1Mk · 2025-10-30

**Soundness:** 2
**Presentation:** 2
**Contribution:** 3
**Rating:** 4
**Confidence:** 5

**Summary:**

This paper proposes a new neuro-symbolic approach for enhancing the logical commonsense reasoning capabilities of LLMs and suggests a new way for LLM’s integration with logic solvers. In the approach proposed in this paper, the LLM iteratively provides unstated commonsense clauses to a logic solver, which is guided by feedback from the solver in the form of the SAT problem backbone. This approach allows the system to perform abductive reasoning, filling in missing background facts while keeping the search tractable. Overall, this work aims to contribute to leveraging the benefits of existing neural and symbolic methods  to tackle commonsense logical reasoning problems.

**Strengths:**

1- The paper is easy to follow and is well-presented (modulo some issues that I point out in the weaknesses). The worked example presented in Section 5.2 and the methodology overview in Figure 3 faciliate understanding of the work.

2- The proposed methodology is novel and insightful. I think the general idea of the work in providing a new paradigm of interaction between an LLM and a symbolic solver is interesting. Existing approaches either initiate reasoning from the LLM and delegate theorem proving to a solver, mimic inference rules using an LLM, or propose methodologies to leverage the LLM to undergo a rigorous reasoning process while leveraging its commonsense. This paper introduces interactions between the LLM and the solver which I find novel and interesting.

3- The topic of focus, commonsense logical reasoning of LLMs is a quite an important topic with numerous practical applications. I think the idea of proposing novel frameworks for LLM interaction with formal reasoners can be be impactful by reducing reasoning errors of LLMs, but the paper's evaluation needs to be strengthened to validate this effect more properly.

**Weaknesses:**

1- There are several statements in Section 3 which I think are vague, inaccurate, or wrong:

- Line 141: “Propositional logic is a logical system that involves propositions about variables.”

   Propositional logic does *not* involve “propositions about variables.” It is a logic of propositions themselves, and variables are symbols for propositions, not objects that propositions are “about.”

 - “A proposition, such as A → ¬B, is some statement about literals”

     Propositions aren’t “about literals”. They’re built from literals (or propositional variables) using logical connectives.

- “We assume that ¬(P ∧ C → ⊥), that is that the premises P are not contradictory with commonsense.”

    The correct way to show consistency is “P∧C⊬⊥”.

-  “A predicate is a function, such as MotherOf (x, y)”

    A predicate represents a property or relation that can be true or false depending on its arguments. Whereas FOL functions point to a particular object in the domain as their output. For example, MotherOf (x, y) is a predicate which can be true or false, but MotherOf (x) is a function that returns a specific object y, i.e., MotherOf (x) = y.

- “∀(x, y)MotherOf (x, y) → ¬Male(x)”

    This is a very unusual syntax. In standard FOL, you either write ∀x ∀y (MotherOf(x, y) → ¬Male(x)) or ∀x ∀y [MotherOf(x, y) → ¬Male(x)].

- Line 190: “First-order logic problems…”

   I encourage the authors to use the conventional terms “grounding” or “instantiation”.

- Line 214: we first try to solve the problem using the SAT Solver (sat_solve) to test whether (P ∧ C) = P ⊢ Q or ¬Q

   what does (P ∧ C) = P ⊢ Q or ¬Q mean? I think you’re just trying to show whether (P ∧ C) ⊢ Q or (P ∧ C) ⊢ ¬Q.

2- The use of self-consistency as one of the ways ARGOS can come up with the final answer is questionable. At the end of the day, self-consistency is relying on the LLM to do the reasoning, but the reason why people leverage or combine symbolic theorem proving with the LLM reasoning is because LLMs alone may make errors in their reasoning or generate hallucinated answers.
The experiments section only reports accuracy of the final answers, whereas in LLM reasoning works such as [1], the correctness of the reasoning process is also critical. Specifically, I think this metric can be insightful to see whether the correctness of reasoning process for answers provided by self-consistency mechanism of ARGOS is also improved or not. This can be a useful complement to the results in Figure 5.

3- There are some typos in the text such as line 81.

[1] Kazemi, Mehran, et al. "LAMBADA: Backward Chaining for Automated Reasoning in Natural Language." Proceedings of the 61st Annual Meeting of the Association for Computational Linguistics (Volume 1: Long Papers). 2023.

**Questions:**

1- In Line 228, the expression rankB(L) = #{L′ ∈ B | L′ has an entity in common with L} is written. Is this intended to be score(L)? Why would the rank of each literal be the number of existing literals that share the same variable? Regardless, the rationale for this approach is unclear to me. The only explanation provided is “which gives a measure of relevance of the literal to the problem” which is vague. I understand space limitations in the main paper, but I strongly suggest you explain the rationale near algorithm 2 in the appendix.

2- The methodology proposed in this work for leveraging the LLM’s commonsense knowledge is restricted to generating literals that can be deduced from the existing literals in the backbone. While this approach is in agreement with the way existing logical reasoning datasets are formed, I don’t think it is general enough for all practical applications. For example, a commonsense rule can be generated using only one literal from the backbone (e.g., ∀x Car(x) → Vehicle(x)), so why are pairs of literals necessary in the proposed approach?

3- Aside from being limited, I think the commonsense rule generation process in this work is also inefficient. Every pair of literals is presented to the LLM, and the scoring mechanism explained in appendix D4 is used to filter irrelevant ones. Some works cited in the paper such as LAMBADA and LLM TRes take a goal-driven approach to only generate rules that can contribute to solving the problem. Why isn’t a similar approach taken in ARGOS?


4- In appendix D3, why are the scoring propositions approaches different across datasets? I suggest a clarifying sentence to explain the rationale. I also appreciate the running example in section 5.2 which facilitates understanding.

5- In line 202, it’s stated that: “Four annotated examples are provided, intended for few-shot prompting.“
How are these few-shot examples chosen? Do they differ per-dataset? Are they chosen in a way that there is no risk of revealing the answer to the LLM?

6- What is the rationale for reducing γ at each iteration? By reducing γ, you are making the method more lenient, accepting answers even if there is less consistency in LLM generations. Doesn’t this approach reduce the rigor of reasoning as the algorithm proceeds?

7- Self-consistency is a key component of ARGOS and in fact one of the ways by which ARGOS generates its answers. As I mentioned in my earlier comments, using the LLM to generate the final answer might be sub-optimal, potentially generating hallucinated answers. Figure 5 nicely provides insight about how ARGOS improves accuracy of self-consistency responses, but I think the paper’s analysis also requires reporting the accuracy of reasons for all methods, at least for one dataset. Also, I think a study in which the self-consistency module of ARGOS is ablated should be provided, at least for one dataset.

8- Regarding RQ1, two mechanisms are used in ARGOS for scoring in the thresholding calculation. Are they both necessary? An additional ablation would be helpful. I also appreciate the honest discussion of limitations in RQ2.

9- I find the discussion in section 6.2 is questionable. Logical translation is a core part of ARGOS, not an orthogonal one and assuming having a correct propositional translation for real applications is a very big assumption to make. I think experiments on at least one dataset is needed without filtering the failures to shed light on how critical this step is to the framework. Using a more powerful LLM than the ones used for generating the commonsense rules is acceptable if it’s a major bottleneck, but having an experiment using the same LLM that ARGOS uses for reasoning is also quite beneficial. A proposed method isn’t required to beat all baselines on all tasks, but the reader must know the strengths and limitations of the proposed method.

---

> ### Author Response · Authors · 2025-11-21
> **Rebuttal to Initial Review (part 1)**
>
> ### Weaknesses
>
> **w1.** There are several statements in Section 3 which I think are vague, inaccurate, or wrong:
>
> **Answer:**
> *We thank the reviewer for this detailed criticism and guidance. We have addressed each of these, and other cases in which similar statements were made, in the text. We have re-worked the problematic logical notations and improper use of logical notions. Please see the updated paper PDF for the changes, in which changes are indicated with colored text. The relevant changes are in the Background section, in the Methodology section at lines 225 and 230, and Proposition 1 at line 708. We provide a more precise definition of propositional logic ("Propositonal logic is a logical system built around propositions, which are [...]", replace the notion of "variables not mentioned" with "propositions not instantiated", we re-write the definition of logical consistency as $P\land C \not \vdash \bot$, and have re-written the expression at line 225 to be clearer. We have also re-written Proposition 1 to be clearer and better-stated.*
>
> **w2.** The use of self-consistency as one of the ways ARGOS can come up with the final answer is questionable. At the end of the day, self-consistency is relying on the LLM to do the reasoning, but the reason why people leverage or combine symbolic theorem proving with the LLM reasoning is because LLMs alone may make errors in their reasoning or generate hallucinated answers.
>
> **Answer:**
> *We thank the reviewer for raising this weakness. We have added discussion of this to the Limitations section at line 537. However, while it is true that LLM-reasoning can be unfaithful or hallucinatory, our approach takes steps to limit this phenomenon's impact on predictions. Broadly, the literature often references self-consistency as a check for unfaithful or hallucinatory reasoning. For an example, see [1], which checks consistency across LLM generations to identify cases of hallucination. Because different hallucinations will lead to different answers, a problem which incites hallucination in the LLM will naturally see more variety in the sampled COT answers and therefore will be lower confidence according to an SC score. Also, by adding information which the LLM would otherwise be required to find during the reasoning process, we decrease the number of steps required to solve the problem. This inherently makes the problem easier. When the problem is easy enough that the LLM is high-confidence, we can expect the LLM to more reliably solve the problem in a faithful way. Besides confidence and simplicity, the mere fact that the required reasoning process now requires fewer steps leaves the LLM less opportunity to hallucinate before answering the query. Relying strictly on the symbolic solver makes the method brittle to symbolic inconsistencies during clause generation, whereas the LLM is naturally robust to this. Despite this, we acknowledge that relying on self-consistency certainly will lead to cases of unfaithful reasoning.*
>
> [1] Cheng, Furui, et al. "Relic: Investigating large language model responses using self-consistency." Proceedings of the 2024 CHI conference on human factors in computing systems. 2024.
>
> **w3.** The experiments section only reports accuracy of the final answers, whereas in LLM reasoning works such as [1], the correctness of the reasoning process is also critical. Specifically, I think this metric can be insightful to see whether the correctness of reasoning process for answers provided by self-consistency mechanism of ARGOS is also improved or not. This can be a useful complement to the results in Figure 5.
>
> **Answer:**
> *We thank the reviewer for this insightful suggestion. We have done manual evaluation of faithfulness of COT and ARGOS, modelled after the experiment done in [1]. We found that, on FOLIO problems for which the final COT answer is correct, Llama 8B generates faithful COT reasoning processes 72% of the time. We found that, on FOLIO problems for which the final ARGOS answer is correct, ARGOS generates faithful COT reasoning processes 85% of the time. This shows that in general ARGOS-L8B is more faithful than pure-COT based methods (i.e. COT, SC). We have added a description of these results, as well as a more in-depth explanation of experimental procedure and analysis, in the Appendix at line 782. Additionally, we have added reference to the new Appendix at line 478.*
>
> **w4.** There are some typos in the text such as line 83.
>
> **Answer:**
> *Thank you, we have addressed this*
>
> [1] Kazemi, Mehran, et al. "LAMBADA: Backward Chaining for Automated Reasoning in Natural Language." Proceedings of the 61st Annual Meeting of the Association for Computational Linguistics (Volume 1: Long Papers). 2023.

---

> ### Author Response · Authors · 2025-11-21
> **Rebuttal to Initial Review (part 2)**
>
> ### Questions
>
> **Q1.** In Line 228, the expression rankB(L) = #\{L' $\in$ B | L' has an entity in common with L\} is written. Is this intended to be score(L)? Why would the rank of each literal be the number of existing literals that share the same variable? Regardless, the rationale for this approach is unclear to me. The only explanation provided is "which gives a measure of relevance of the literal to the problem" which is vague. I understand space limitations in the main paper, but I strongly suggest you explain the rationale near algorithm 2 in the appendix.
>
> **Answer:**
> *We thank the reviewer for pointing out this gap in our explanation. Intuitively, the FOL variable about which we know the most from the given problem context is likely to be relevant to the problem. This is because while problems often include irrelevant or distractor information, it is unlikely that the majority of it will be irrelevant to the query. Take the CLUTRR problem structure, for example. If we are given six relations about John and only one about Jane, it is more likely that John will be a link in the reasoning chain than Jane and so we should prioritize finding new information about him. We have added text to this effect in the methodology section at line 248.*
>
> **Q2.** The methodology proposed in this work for leveraging the LLM's commonsense knowledge is restricted to generating literals that can be deduced from the existing literals in the backbone. While this approach is in agreement with the way existing logical reasoning datasets are formed, I don't think it is general enough for all practical applications. For example, a commonsense rule can be generated using only one literal from the backbone (e.g., $\forall$x Car(x) $\rightarrow$ Vehicle(x)), so why are pairs of literals necessary in the proposed approach?
>
> **Answer:**
>
>
> *We thank the reviewer for raising this gap in the explanation for our method. Pairs of literals are not strictly necessary for the proposed approach. While we generate propositions of the form $L_1 \land L_2 \rightarrow L_{right}$, if $L_1$ and $L_2$ are the same literal, the resulting rule is effectively $L_1 \rightarrow L_{right}$, a single-backbone-literal proposition. Similarly, if the literals $L_1$ and $L_2$ are both chosen to be empty, we get $\vdash L_{right}$, a zero-backbone-literal rule. We have added this detail to the explanation of our method at line 242.*
>
> **Q3.** Aside from being limited, I think the commonsense rule generation process in this work is also inefficient. Every pair of literals is presented to the LLM, and the scoring mechanism explained in appendix D4 is used to filter irrelevant ones.
>
> **Answer:**
> *We thank the reviewer for this comment, and we do acknowledge that it is inefficient. We have not prioritized identifying a more efficient approach because the rule-generation calls are a very small fraction of the overall cost of the method. The generative call in the rule generative process generates at most 25 tokens, compared to 300 for COT calls. Since in our implementation we generate one new rule every 5 COTs, a 90% rejection rate would only result in an 18% increase in cost (since each proposed rule costs 25 tokens + $2\times1$ scoring token). In practice, our acceptance criteria are lax: we use a low acceptance threshold, typically 0.3, and the characteristics we score for are themselves simple: general contextuality and commonsensicality, both of which the generation prompt itself encourages. We typically see a rejection rate of 50-80%. An 80% rejection rate would mean that out of 5 rules, 4 are rejected, and would raise the cost by 9%, less than the cost of two COTs. A 50% rejection rate would raise the cost by 0.036%, less than the cost of one COT. In fact, we have added text in the Appendix at line 725 discussing that SC takes 2 hours longer than ARGOS on FOLIO, because FOLIO uses fewer COTs on average than SC, eclipsing the additional cost incurred by ARGOS from scoring and rule-generation calls.*

---

> ### Author Response · Authors · 2025-11-21
> **Rebuttal to Initial Review (part 3)**
>
> **Q3.** Aside from being limited, I think the commonsense rule generation process in this work is also inefficient. Every pair of literals is presented to the LLM, and the scoring mechanism explained in appendix D4 is used to filter irrelevant ones.
>
> **Answer:**
> *We thank the reviewer for this comment, and we do acknowledge that it is inefficient. We have not prioritized identifying a more efficient approach because the rule-generation calls are a very small fraction of the overall cost of the method. The generative call in the rule generative process generates at most 25 tokens, compared to 300 for COT calls. Since in our implementation we generate one new rule every 5 COTs, a 90% rejection rate would only result in an 18% increase in cost (since each proposed rule costs 25 tokens + $2\times1$ scoring token). In practice, our acceptance criteria are lax: we use a low acceptance threshold, typically 0.3, and the characteristics we score for are themselves simple: general contextuality and commonsensicality, both of which the generation prompt itself encourages. We typically see a rejection rate of 50-80%. An 80% rejection rate would mean that out of 5 rules, 4 are rejected, and would raise the cost by 9%, less than the cost of two COTs. A 50% rejection rate would raise the cost by 0.036%, less than the cost of one COT. In fact, we have added text in the Appendix at line 725 discussing that SC takes 2 hours longer than ARGOS on FOLIO, because FOLIO uses fewer COTs on average than SC, eclipsing the additional cost incurred by ARGOS from scoring and rule-generation calls.*
>
> **Q4.** Some works cited in the paper such as LAMBADA and LLM TRES take a goal-driven approach to only generate rules that can contribute to solving the problem.
>
> **Answer:**
> *We disagree that LLM-TRES takes a goal-driven approach. LLM-TRES uses a language model to calculate an entailment score for each potential antecedent/implicand pair (note that LLM-TRES restricts itself to single-literal antecedents) and orders the resulting propositions by score. LLM-Tres does not only generate rules which can contribute to solving the problem. Since a computation limit is imposed on LLM-Tres, and high-score pairs are prioritized, this is effectively a scheme for filtering out low-entailment-score pairs. This filtering scheme, at a high level, is not very different from ARGOS', and so to distinguish LLM-Tres as being goal-driven but ARGOS as being inefficient is inaccurate. We also disagree that LAMBADA takes a goal-driven approach to generate. This is because LAMBADA does not generate new rules, it selects from existing rules in its Rule Selection stage.*
>
> **Q5.** Why isn't a similar approach taken in ARGOS as in LAMBADA and LLM-TRES to generate new rules?
>
> **Answer:**
> *A similar approach is not taken in ARGOS as in LLM-Tres because while LLM-Tres succeeds well in minimizing rule-generation cost by aggressively restricting the space such that autoregressive generation is not required, this restriction means that it struggles to capture the full space of potentially necessary information. Firstly, it is restricted to only single-literal antecedents. Secondly, implicands can only be made up of FOL predicates and variables which were already insantiated by the problem context. While these restrictions lead to cheap and strong performance on simpler problems, complex problems tend to require multi-literal reasoning steps. These criticisms, and associated empirical results demonstrating poor performance on complex reasoning tasks, motivated us to avoid the generation framework of LLM-TRES. This is reflected in the text at line 139, where we criticise LLM-Tres in the Related Work section for such restrictions.*
>
> **Q6.** In appendix D3, why are the scoring propositions approaches different across datasets? I suggest a clarifying sentence to explain the rationale.
>
> **Answer:**
> *The scoring approaches are different across datasets because CLUTRR's more consistent linguistic structure allows for very straightforward prompt formats. For more linguistically complex datasets, however, more robust prompt formatting is required. For example, asking if a rule seems contradictory is more robust in situations where a rule is ambiguous without context (i.e., student(Rina) implies like\_coffee(Rina), vs. Mom(Hannah, Sam) and Sibling(Sam,Mary) implies Mom(Hanna, Mary)). We make brief reference to the structural difference as a rationale on l. 928. We have added this more detailed discussion to the text at line 928.*

---

> ### Author Response · Authors · 2025-11-21
> **Rebuttal to Initial Review (part 4)**
>
> **Q8.** What is the rationale for reducing $\gamma$ at each iteration? By reducing $\gamma$, you are making the method more lenient, accepting answers even if there is less consistency in LLM generations. Doesn't this approach reduce the rigor of reasoning as the algorithm proceeds?
>
> **Answer:**
> *The iterative reduction of $\gamma$ allows us to limit computation. Mention of this has been added at line 233. The alternative to gradually reducing the threshold would be to, at a certain iteration count, simply terminate ARGOS and either take a best-guess, or to reject the problem as unsolved. Gradual reduction is both more cost sensitive than the best-guess approach and more natural to the experimental setting than rejection, while still allowing us to impose a hard-limit on iterations (when threshold confidence is 0.5 a decision is guaranteed).*
>
> **Q9.** Self-consistency is a key component of ARGOS and in fact one of the ways by which ARGOS generates its answers. As I mentioned in my earlier comments, using the LLM to generate the final answer might be sub-optimal, potentially generating hallucinated answers. Figure 5 nicely provides insight about how ARGOS improves accuracy of self-consistency responses, but I think the paper's analysis also requires reporting the accuracy of reasons for all methods, at least for one dataset.
>
> **Answer [duplicated from W3]:**
> *We thank the reviewer for this insightful suggestion. We have done manual evaluation of faithfulness of COT and ARGOS, modelled after the experiment done in [1]. We found that, on FOLIO problems for which the final COT answer is correct, Llama 8B generates faithful COT reasoning processes 72% of the time. We found that, on FOLIO problems for which the final ARGOS answer is correct, ARGOS generates faithful COT reasoning processes 85% of the time. This shows that in general ARGOS-L8B is more faithful than pure-COT based methods (i.e. COT, SC). We have added a description of these results, as well as a more in-depth explanation of experimental procedure and analysis, in the Appendix at line 782. Additionally, we have added reference to the new Appendix at line 478.*
>
> **Q10.** Also, I think a study in which the self-consistency module of ARGOS is ablated should be provided, at least for one dataset.
>
> **Answer:**
> *We thank the reviewer for this suggestion. We have included the result in the Appendix at line 864, with reference in the main text at line 433. We ablate the SC-Solver option from ARGOS, attempting only symbolic solution and implementing a timeout. We find that removing the SC option only somewhat hurts ARGOS performance on CLUTRR, but has a catastrophic effect on FOLIO. This is not at all surprising, since CLUTRR is far simpler and more logically structured than FOLIO. Interestingly, we find that LLM-Tres performs slightly better than ARGOS-Symbolic on FOLIO. On investigation, we found that propositions with a single-literal antecedent were generally enough to solve FOLIO problems, and so LLM-Tres' methodological restriction to such rules benefits it, whereas in CLUTRR where nearly all necessary propositions have two literals in the antecedent, LLM-Tres is reduced to guessing. This massive performance gap which we see on ARGOS in the more linguistic and more logical datasets illustrates the need for balanced, neuro-symbolic approaches.*
>
> |                    | FOLIO 8B | CLUTRR 8B |
> |--------------------|----------|-----------|
> | ARGOS-Symbolic     | 59%      | 72%       |
> | ARGOS              | 81%      | 76%       |
> | SC                 | 71%      | 69%       |
> | LLM-Tres           | 63%      | 51%       |
>
> **Table:** Ablating the SC-solver on ARGOS. ARGOS-Symbolic denotes the ablated version of ARGOS.

---

> ### Author Response · Authors · 2025-11-21
> **Rebuttal to Initial Review (part 5)**
>
> **Q11.** Regarding RQ1, two mechanisms are used in ARGOS for scoring in the thresholding calculation. Are they both necessary? An additional ablation would be helpful. I also appreciate the honest discussion of limitations in RQ2.
>
> **Answer:**
> *We thank the reviewer for this suggestion. We have added this result to the Appendix at line 894. In the table, we ablate separately the commonsense-ness and contextual-ness scores. We find that both are necessary. This makes sense, since while they may have some overlap in terms of the rules which they serve to reject (for example, gibberish output would be neither commonsensical nor contextually relevant), it is easily conceivable that the sets of proposed propositions which they reject correctly is not totally overlapping. Also, we have conducted a human review of 50 ARGOS-proposed rules and their corresponding scores, for Llama 8B on FOLIO. We found that, by classifying with the decision rule of thresholding at 0.3 for commonsense-ness and contextual-ness separately, the commonsense binary classification is correct 76% of the time, and the contextualness classification accuracy is correct 91% of the time. We find that 21% of the proposed rules are only rejected by one of the two decision thresholds, indicating that both are necessary 21% of the time. We have added this to the appendix at line 990.*
>
> |                    | FOLIO 8B |
> |--------------------|----------|
> | No Commonsense     | 79%      |
> | No Contextual      | 80%      |
> | Full ARGOS         | 81%      |
>
> **Table:** Ablating individual score thresholds
>
>
> **Q12.** I find the discussion in section 6.2 is questionable. Logical translation is a core part of ARGOS, not an orthogonal one and assuming having a correct propositional translation for real applications is a very big assumption to make.
>
> **Answer:**
> *The reviewer makes a good point. While not orthogonal, a deployed reasoning system would likely consist of a translation module and an ARGOS-like proof-planning module. These modules would likely be separate, since it is unclear that there would be any benefit to merging the two as they each address distinct tasks. While an excellent proof-planner may be able to provide feedback to the translator, or to identify cases of mistranslation, this is work for the future and out of scope for this work, as we focus on the proof planning itself. In development of a method, it is critically important to ensure that it performs well given the correct input. At the current state of the art, many existing methods, as we show in our experimental results, fail in a substantial percentage of cases even when the input is correct, because the reasoning is challenging. So, focusing on improving reasoning even when the input is correct is very important. Despite this, we do empirically evaluate performance when logical translation is imperfect. In Section 5.2, we do so by reporting results on an un-filtered FOLIO set translated by Claude Opus, finding that on the unfiltered data ARGOS still outperforms the baselines, with only a 2-percent accuracy reduction. We have modified the text at line 482 and 509 to highlight the fact that we verify our claim with empirical result, and to modify our claim of orthogonality to better reflect our argument. To strengthen this argument, we have also repeated this experiment on QUAIL which is much less logically structured and more prone to mistranslation, finding that ARGOS goes from 82% accuracy to 73%. While this is a larger drop in performance, ARGOS remains the best-performing method, with SC at 68% accuracy. These results have been added to the paper at line 514.*
>
> **Q13.** I think experiments on at least one dataset is needed without filtering the failures to shed light on how critical this step is to the framework. Using a more powerful LLM than the ones used for generating the commonsense rules is acceptable if it's a major bottleneck, but having an experiment using the same LLM that ARGOS uses for reasoning is also quite beneficial.
>
> **Answer:**
> *We presented this result in Section 5.2 on ARGOS-L8B and on the FOLIO dataset. We use Claude Opus to translate problems, and Llama 8B as the reasoning backbone. We realize that this experiment was somewhat hidden in the original paper, and have modified the text at line 482 to highlight this experiment.*

---

> > ### Comment · Reviewer_L1Mk · 2025-11-24
> >
> > I appreciate the authors’ efforts in the rebuttal. The new experiments and edits have improved the paper. I still have a few remaining concerns:
> >
> > 1- Regading Q1: I still think the definition rankB(L) = #{$L' \in B | L'$ has an entity in common with L} is not correct. In your example, if six relations are known about John and only one about Jane, would John really be ranked sixth and Jane first? It seems more like you compute a score using this formula and then rank the literals by that score. If this is the case, please make it clear.
> >
> > 2- Regading Q2: are you considering all subsets of literals up to size 2? Does this mean you assume there is no commonsense rule whose body has 3 or more literals?
> >
> > 3- Regading Q4: LAMBADA and LLM-TRES both use backward chaining, which is goal-driven (they start from the query and work backward, using only rules that help prove or disprove it). Forward chaining, which you use, starts from the known facts and applies rules until the goal can be proven or refuted. Forward chaining is also a valid and widely-used approach, but if you intentionally chose it, I think the paper should explain why. In your response to Q5, you mention that LLM-TRES struggles to capture the full space of information it needs. Could you explain this more? Is this the main reason you chose forward chaining? Can you provide an example for the case that the backward chaining approach in LLM-TRES cannot generate the required commonsense rule that serves proving the goal, while ARGOS can?

---

> > > ### Author Response · Authors · 2025-11-25
> > > **Part 2**
> > >
> > > 6. Can you provide an example for the case that the backward chaining approach in LLM-TRES cannot generate the required commonsense rule that serves proving the goal, while ARGOS can?
> > >
> > >    **Answer:**
> > >    *To clarify, backwards chaining has no intrinsic limitation on the rules which can be generated. The limitation comes from the fact that by design, LLM-Tres can not generate any multi-literal antecedents, and it can not instantiate new literals. So, Mother(Sarah, John) ∧ Daughter(Nancy, John) → GrandDaughter(Nancy, Sarah), for example, is outside LLM-Tres's generation space. If LLM-Tres were adapted to allow for up to 2 literals in the antecedent, we can discuss how many rules would need to be tested by LLM-Tres in order to build the full priority queue at each iteration. Say that there are 5 family members in the given family-relationship problem. Since there are 24 family relations, and 5 × 4 pairs of individuals, there are 24 × 20 = 480 literals in the problem. Since antecedents can be formed by any two literals, there are 480 × 479 = 229,920 rules to be tested. LLM-Tres is designed based on the assumption that multiple separate proofs can be tested. If a single complete scoring of the literals on a single iteration is impossibly expensive, the method will clearly break down.*

---

> > > > ### Comment · Reviewer_L1Mk · 2025-11-26
> > > >
> > > > Thank you for your response. I think the paper is in a much better shape now, and I have updated my score accordingly.

---

> ### Author Response · Authors · 2025-11-25
> **Part 1**
>
> 1. Regarding Q1: I still think the definition rankB(L) = #{L' ∈ B | L' has an entity in common with L} is not correct. In your example, if six relations are known about John and only one about Jane, would John really be ranked sixth and Jane first? It seems more like you compute a score using this formula and then rank the literals by that score. If this is the case, please make it clear.
>
>    **Answer:**
>    *We thank the reviewer. Indeed, the expression should be written  scoreB(L) = #{L' ∈ B | L' has an entity in common with L}, so that we take the literals with the greatest scores first. We have updated the text at line 246.*
>
> 2. Regarding Q2: are you considering all subsets of literals up to size 2? Does this mean you assume there is no commonsense rule whose body has 3 or more literals?
>
>    **Answer:**
>    *Yes, we consider all subsets of literals up to size 2 as antecedents. This is potentially a weakness of the method, and we have already added it as a limitation at line 532, in response to reviewer Kqo3. However, it is worth noting the well-known result that any logic expressed in CNF form is transformable to 3-CNF form such that the newly defined SAT problem is equisatisfiable. That is, all CNF-form logic problems can be expressed with exactly 3-literal clauses (rules with antecedents made up of 2 literals) (Aho et al. 1974, pp. 384). More concretely, we argue that commonsense rules with more than three atoms in the body of the clause (the antecedent) are often more commonsensical when decomposed into smaller sub-rules, meaning that commonsense is fully (and more naturally) described by only few-antecedent rules. For example, "Since it's winter and the fox is warm-blooded then to survive the winter the fox becomes white" can be described either as one long rule: winter(season) ∧ must_survive(fox) ∧ warm_blooded(fox) → white(fox), or as several sub-rules: (i) "Winter is cold" (winter(season) → cold(weather). (ii) "To survive, warm-blooded animals must stay warm" (must_survive(fox) ∧ warm_blooded(fox) → must_stay_warm(fox)) (iii) To stay warm in the cold, the fox turns white (cold(weather) ∧ must_stay_warm(fox) → white(fox)). Each of these sub-rules are more obvious/common-sensical than the longer 3-atom-head alternative.*
>
> 3. Regarding Q4: LAMBADA and LLM-TRES both use backward chaining, which is goal-driven (they start from the query and work backward, using only rules that help prove or disprove it). Forward chaining, which you use, starts from the known facts and applies rules until the goal can be proven or refuted. Forward chaining is also a valid and widely-used approach, but if you intentionally chose it, I think the paper should explain why. In your response to Q5, you mention that LLM-TRES struggles to capture the full space of information it needs. Could you explain this more?
>
>    **Answer:**
>    *The reviewer makes a good point. LLM-Tres does indeed search for resolvent clauses starting at the query (i.e., backward chaining), which is certainly a goal-oriented approach. In our previous response, we took a more general meaning that LLM-Tres still tests all clauses given the space, an approach which in itself is not goal-oriented in a broader sense. The reason that LLM-Tres struggles to capture the full space is that the space of clauses from which it can select as proposed resolvent clauses is limited to be two-literal clauses (one antecedent, one implicand), such that both literals are instantiated by the initial problem. This is necessary for the LLM-Tres method because it effectively guarantees that the resolution tree which is being built starting at the query-literal will eventually join with the partial resolution tree given by the input within a computational budget. If this restriction is not imposed, then the chosen antecedent can be about a new relation or entity on which we have no information, unless successive steps eventually tie this information to the givens of the problem.*
>
> 4. Is this the main reason you chose forward chaining?
>
>    **Answer:**
>    *We chose forward-chaining primarily for its simplicity. We found that, evidenced by methods such as LAMBADA, that backwards-chaining with the LLM requires multi-step and multi-prompt algorithms, whereas forward-chaining with the LLM requires only a single COT prompt. Additionally, we found that back-chaining methods tended to show reliability issues, stemming from essentially an LLM-guided recursive search of the proof. **We've added this explanation for our choice of forward-chaining to the text at line 254. We've also added to the limitations that a potential future direction of research may be to investigate backwards-chaining approaches for abductive reasoning, at line 539.***

---

### Official Review · Reviewer_mBHM · 2025-10-31

**Soundness:** 3
**Presentation:** 3
**Contribution:** 3
**Rating:** 4
**Confidence:** 3

**Summary:**

The paper proposes a neuro-symbolic framework called ARGOS to improve commonsense reasoning. This framework addresses the inability of logic solvers to handle missing commonsense facts by using an LLM to iteratively provide new commonsense propositions. An interesting contribution is the use of feedback from the symbolic SAT solver itself to guide the LLM's search for relevant facts. This allows ARGOS to search a larger space of potential facts, including new variables not present in the original problem. The framework also uses the LLM to score the generated facts for commonsense and relevance before adding them . The authors show that this approach improves performance on three abductive reasoning datasets.

**Strengths:**

- The proposed method provides an intuitive framework for combining the strengths of symbolic solvers and LLMs.
- The use of the SAT solver's backbone to guide the generation of new commonsense facts is novel.
- The empirical results are strong and show consistent improvements over existing neural and symbolic baselines on 3 datasets.
- The ablation studies show the value of the two main contributions, ie, backbone-guided search and score-based thresholding.

**Weaknesses:**

- The tasks/datasets used are not practically relevant and lack real-world applicability. In addition, the paper relies on modified versions of existing datasets (ProntoQA, CLUTRR, FOLIO) to create an abductive setting, which means the evaluation is on a somewhat artificial task.
- The method requires logit-level access to score generated clauses for commonsense and relevance, which may not always be accessible for closed-source models.
- The main experiments assume a perfect logical translation from text, as failed translations were filtered out. But this ignores the issue of imperfect translation, which could be a bottleneck for this method and neuro-symbolic systems in general.
- The method relies on an LLM itself to score its own generated clauses for commonsense and relevance. The reliability of this LLM-as-a-judge component is not validated against human-annotated scores.
- There is a lack of examples accompanying the error analysis in the paper, showing failure cases of ARGOS.
- The paper does not report the latency of all approaches compared in the main results. This is important because it would seem to me that ARGOS likely takes much higher computation time.

**Questions:**

- Since ARGOS depends on the LLM's ability to reliably score commonsense and relevance, did you do any human analysis to verify that the LLM-generated scores are calibrated & accurate?
- How often does ARGOS introduced an unseen variable that is important for solving the problem?

---

> ### Author Response · Authors · 2025-11-21
> **Rebuttal to Initial Review (part 1)**
>
> ### Weaknesses
>
> **w1.** The tasks/datasets used are not practically relevant and lack real-world applicability.
>
> **Answer:**
> *In the summary section of the review, and in the strengths section, the reviewer has referred to our use of three datasets ("The authors show that this approach improves performance on three abductive reasoning datasets"). In fact, we use 6 datasets, reporting our method's accuracy on each of the 6. With that said, while the datasets are standard benchmarks for logical reasoning and must be included in academic study, we agree with the reviewer that datasets FOLIO, CLUTRR, and ProntoQA are not practically relevant and are not real-world applicable because of their strong logical structure. For this reason, the three additional datasets, QUAIL, ESNLI and CosmosQA, were all chosen because they are non-logical in structure and serve to test our method's generalizability to more natural forms of questions which real users may have. In particular, both QUAIL and CosmosQA are reading comprehension-based problems, which have clear real-world applicability as many people today use AI tools for document summarization. We have added this discussion to the benchmarks subsection at line 338, and thank the reviewer for raising this point.*
>
> **w2.** In addition, the paper relies on modified versions of existing datasets (ProntoQA, CLUTRR, FOLIO) to create an abductive setting, which means the evaluation is on a somewhat artificial task.
>
> **Answer:**
> *It is true that we modify the datasets to create a modified version of the task. We thank the reviewer for highlighting this, and have added it to the limitations section at line 537. With that said, we argue that the original tasks represented by these datasets are themselves artificial. For example, in CLUTRR, the full list of abstract relational reasoning rules are provided to the logic solver. In practice, when users have even strongly logically-structured problems to ask the AI system, the user cannot be expected to provide commonsense information, as the LLM is expected to know this. In FOLIO and in ProntoQA the same can be said: the user cannot be expected to always use explicit and clear language or to specify, for example, that all cats are mammals.*
>
> **w3.** The method requires logit-level access to score generated clauses for commonsense and relevance, which may not always be accessible for closed-source models.
>
> **Answer:**
> *This is true, and we acknowledge that this is currently a limitation of our method. We could imagine extending our work by changing the prompts for scoring the clauses, e.g., by directly asking the LLM to provide a verbal score from 0 to 1. Shifting to a verbal scoring system, however, would still require some meaningful design iteration and experimentation. We have added discussion of this point to the limitations section, at line 535.*
>
> **w4.** The main experiments assume a perfect logical translation from text, as failed translations were filtered out. This ignores the issue of imperfect translation, which could be a bottleneck for this method and neuro-symbolic systems in general.
>
> **Answer:**
> *It is true that our main experiments assume a perfect logical translation from text. This was a deliberate choice, as we wanted experimentation to focus on the reasoning capabilities of the methods rather than their ability to recover from translation errors. However, in section 6.2 (now 5.2, in the updated draft) of the paper, we claim that translation error is likely to be of low impact compared to reasoning error, given that translation is a far more natural and common task for LLMs than reasoning. We also tested this claim by repeating the main experiment with SC and ARGOS using Llama 8B on an unfiltered FOLIO, finding that ARGOS only sees a marginal drop in performance, from 80% to 78% accuracy. These results were reported in Section 5.2, at line 482. Despite this, ARGOS remained the outperforming method, with the next best method (SC) performing at 71%. To strengthen this argument, we have also repeated this experiment on QUAIL which is much less logically structured and more prone to mistranslation, finding that ARGOS goes from 82% accuracy to 73%. While this is a larger drop in performance, ARGOS remains the best-performing method, with SC at 68% accuracy. These results have been added to the paper at line 514.*

---

> ### Author Response · Authors · 2025-11-21
> **Rebuttal to Initial Review (part 2)**
>
> **w5.** The method relies on an LLM itself to score its own generated clauses. The reliability of this LLM-as-a-judge component is not validated against human-annotated scores.
>
> **Answer:**
> *We thank the reviewer for this suggestion. We have conducted a human review of 50 ARGOS-proposed rules and their corresponding scores, for Llama 8B on FOLIO. We found that, by classifying with the decision rule of thresholding at 0.3 for commonsense-ness and contextual-ness separately, the commonsense binary classification is correct 76% of the time, and the contextualness classification accuracy is correct 91% of the time. We have added this to the appendix at line 992. With that said, since we use the LLM-scoring module not to evaluate our method, but as a component of our method, measuring whether the scoring component of the method contributes to correct predictions was sufficient validation. We don't really care if there is alignment between human-annotated scores and the LLM-as-a-judge; we just need the LLM to provide useful scores. The validation was therefore focused on whether the scores are useful, not whether they align with human scores. When we ablate the scoring/thresholding mechanism (Table 2), immediately allowing all generated propositions to be added to the problem, we see a drop in performance on FOLIO with Llama 8B from 81% to 79%. If the scoring mechanism were ineffective, we would be rejecting rules arbitrarily/randomly, and its ablation would have no effect on accuracy. So, the fact that its ablation decreases accuracy validates its effectiveness.*
>
> **w6.** There is a lack of examples accompanying the error analysis in the paper, showing failure cases of ARGOS.
>
> **Answer:**
> *In fact, we provide 3 examples of failed ARGOS problems on FOLIO in the Appendix J (line 1184), referenced at line 476 of the main text. We felt that given the limited space in the main text, the inclusion of the figures 5(a) and 5(b), and the inclusion of the ablation study in Table 3, were more important than the analysis of more examples.*
>
> **w7.** The paper does not report the latency of all approaches compared in the main results. This is important as it would seem to me that ARGOS likely takes much higher computation time.
>
> **Answer:**
> *In Table 3, in the appendix at line 756, we describe method latency in terms of number of COTs required per problem on average. We find that ARGOS requires fewer COT calls on average than self-consistency. We also find that despite the fact that ARGOS requires several additional LLM calls, ARGOS sees a lower runtime on FOLIO than self-consistency because these additional calls are very small and cheap relative to complete COT call with few-shot examples in the context.*
>
> ### Questions
>
> **Q1.** Since ARGOS depends on the LLM's ability to reliably score commonsense and relevance, did you do any human analysis to verify that the LLM-generated scores are calibrated and accurate?
>
> **Answer:**
> *We did not conduct human analysis to verify that the LLM-generated scores are calibrated and accurate. When ablating the scoring modules (i.e. accepting all proposed new propositions), we found that ARGOS performance measured on FOLIO drops from 81% to 79%. This serves to validate the scoring component of the method. With that said, we have conducted a human review of 50 ARGOS-proposed rules and their corresponding scores, for Llama 8B on FOLIO. We found that, by classifying with the decision rule of thresholding at 0.3 for commonsense-ness and contextual-ness separately, the commonsense binary classification is correct 76% of the time, and the contextualness classification accuracy is correct 91% of the time. We find that the scores are not in general well-calibrated, but this is not needed for the thresholding that we do. We have added this to the appendix at line 992.*
>
> **Q2.** How often does ARGOS introduce an unseen variable that is important for solving the problem?
>
> **Answer:**
> *We thank the reviewer for this excellent question. We have conducted a new experiment, finding that ARGOS, on Llama 8B, identifies important new variables for 65% of the CLUTRR questions we test on. Details of this experiment have been added under RQ2 of the experimental section, at line 445. We compile single proofs by running a SAT solver on the full CLUTRR problem including the abstract family relational rules, and check how frequently problems have successfully been augmented with information about variables in the proof by ARGOS. The title of RQ2 at line 435 has also been appropriately modified to reflect this.*

---

> ### Author Response · Authors · 2025-11-27
>
> Hi, please let us know if we have answered your concerns to your satisfaction. If we have not, we ask that you please list your remaining concerns, so that we might address them

---

### Official Review · Reviewer_Kqo3 · 2025-10-31

**Soundness:** 2
**Presentation:** 3
**Contribution:** 2
**Rating:** 4
**Confidence:** 3

**Summary:**

The paper presents ARGOS, a neuro-symbolic framework designed to solve logic problems that require abductive reasoning—the ability to infer missing commonsense information. It addresses a well-known gap in existing systems: while symbolic solvers are rigorous, they are brittle and require a complete set of premises, whereas Large Language Models (LLMs) possess vast commonsense knowledge but often fail at complex proof planning.

**Strengths:**

- Clarity: The paper is exceptionally well-written and easy to follow.
- Quality: The experiment is executed to a high standard, both methodologically and empirically.

**Weaknesses:**

- Durability of the Problem Statement Against Frontier Models: The paper's motivation hinges on the inability of LLMs to perform abductive reasoning. I find that SOTA thinking models like Gemini-2.5-pro can solve the paper's motivating "winter fox" example directly via chain-of-thought. This raises the question of whether the proposed method addresses a fundamental limitation or a capability gap in a specific class of models that may soon be obsolete.

- Worst-Case Complexity: The paper reports an average cost of 18.4 COT calls (Table 3), which is reasonable. However, the worst-case cost is unbounded in theory and in practice determined by the number of iterations allowed. For very hard problems requiring many abduction steps, the cost could become prohibitive, as each iteration involves multiple LLM calls (generation, commonsense scoring, relevance scoring) and solver calls. A discussion of the distribution of costs, not just the average, and the performance/cost trade-off would be valuable.

- Inability to Express More Complex Rules: Real-world commonsense often takes more complex forms.  The current llm_generate prompt structure seems hard-coded for the two-antecedent form. The paper would be more complete if it acknowledged this limitation and discussed potential extensions.

**Questions:**

Minor suggestion on paper structure: Section 4 ("PROBLEM STATEMENT") is very concise and could be integrated into the end of Section 3 ("BACKGROUND") to improve the paper's narrative flow.

---

> ### Author Response · Authors · 2025-11-21
> **Rebuttal to Initial Review (part 1)**
>
> ### Weaknesses
>
> **w1.** Durability of the Problem Statement Against Frontier Models: The paper's motivation hinges on the inability of LLMs to perform abductive reasoning. I find that SOTA thinking models like Gemini-2.5-pro can solve the paper's motivating "winter fox" example directly via chain-of-thought. This raises the question of whether the proposed method addresses a fundamental limitation or a capability gap in a specific class of models that may soon be obsolete.
>
> **Answer:**
> *The reviewer makes a good point, and as a result we have added some discussion in the introduction at line 29 discussing the motivation of researching approaches to improve open-sourced models. In truth, we cannot say whether SOTA closed-source models are in fact using simple chain of thought, or more complex algorithmic refinement/search techniques. If these closed-source systems are indeed doing refinement, then investigation into such systems is well-motivated as a way of adapting open-source models towards the state of the art, and as a way of understanding how these closed-source models really work. Clearly, the broad literature values such open-source investigations, as there are hundreds, if not thousands, of papers published in top AI conferences evaluating open-sourced methods on datasets which the closed-source SOTA AI systems can already address reliably. We would argue strongly that it should not be assumed that a problem is satisfactorily solved just because a closed-source commercial product can provide a solution. We also point out that the winter fox problem given as an example is not a part of any evaluation set we used, and is only an illustrative example chosen in particular for its straightforward simplicity.*
>
> **w2.** Worst-Case Complexity: The paper reports an average cost of 18.4 COT calls (Table 3), which is reasonable. However, the worst-case cost is unbounded in theory and in practice determined by the number of iterations allowed.
>
> **Answer:**
> *We thank the reviewer for raising this point. Firstly, since the iterative threshold reduction (which guarantees eventual termination of the algorithm, at worst case when confidence threshold is at 0.5) is an element of our method's design, we can in fact say that worst-case cost of our method is bounded (in theory) by the number of COTS per iterative self-consistency check multiplied by the number of iterations before the threshold hits 0.5 (which is determined by the threshold reduction schedule). Secondly, even if the threshold reduction were considered as external intervention on the method for computational considerations, this criticism of theoretical unboundedness is applicable for any LLM-generation-based reasoning method, since autoregressive generation is in theory unbounded and only restricted by a manually enforced token limit. We have added such discussion at line 234.*
>
> **w3.** For very hard problems requiring many abduction steps, the cost could become prohibitive, as each iteration involves multiple LLM calls (generation, commonsense scoring, relevance scoring) and solver calls.
>
> **Answer:**
> *The reviewer raises an excellent point. Harder problems, which require more proof steps, require more ARGOS iterations. This can be seen in Figure 5(b) of the paper, where problems on which SC does worse (and are therefore harder in our view) are assigned more ARGOS iterations. However, the generation and commonsense/relevance scoring calls are trivial in cost relative to a single COT call. For implicand literal generation, we set the token generation limit to 25. For scoring, token limit is effectively 1, since we softmax the measured probabilities of the Yes and No tokens. In practice, we see that on FOLIO, self-consistency takes longer than ARGOS: SC takes 24 hours and 58 seconds, whereas ARGOS takes 21 hours, 48 minutes and 3 seconds. This is explained by the fact that ARGOS averages fewer than 20 COTs per problem on FOLIO, and that the ARGOS-specific LLM calls are so small as to not contribute meaningfully to the full run-time. These concrete runtimes have been added to the discussion at lines 725-732 arguing that COT-based cost measurement is the most appropriate complexity metric.*

---

> ### Author Response · Authors · 2025-11-21
> **Rebuttal to Initial Review (part 2)**
>
> **w4.** Discussion of the distribution of costs---not just the average---and the performance/cost tradeoff would be valuable.
>
> **Answer:**
> *Thank you for pointing this out. We have added a histogram of costs, and discussion of the observed distribution, to the Appendix at line 779. We observe that for most problems, only 10 COTs are required. This allows ARGOS to exceed the on-average cap for problems requiring more ablation or deeper search (i.e., harder problems). This leads us to consider the cost-performance tradeoff, for which we have added discussion at line 739. To make the discussion more concrete, we compose a new dataset of only the hardest CLUTRR problems, those for which ARGOS takes 8-10 ARGOS steps (the final bar in Figure 5(b)). Testing ARGOS, limited at 20 COTs per-problem (strictly per problem, not on average across problems), we see that its performance drops from 65% to 54% on these problems, whereas SC performs at 40% with these problems. This indicates a clear and steep cost-performance tradeoff for ARGOS, but even with the strict cost limitations ARGOS outperforms SC considerably.*
>
> **w5.** The current llm\textunderscore generate prompt structure seems hard-coded for the two-antecedent form.
>
> **Answer:**
> *We thank the reviewer for pointing out this lack of clarity in the text. The current llm\textunderscore generate prompt structure is not hard-coded for the two antecedent form; it is structured for up to two antecedents. Since we make no specification that the antecedent set is composed of two distinct literals from the backbone, in the case where the two selected antecedent variables are the same backbone variable, this translates to a 1 antecedent rule. We realize this was not clear in the text, and have edited the methodology section at line 241 to clarify this.*
>
> **w6.** Real-world commonsense often takes complex forms. The paper would be more complete if it acknowledged this limitation and discussed potential extensions.
>
> **Answer:**
> *We thank the reviewer for this suggestion. We have added to the limitations section discussion of the restrictions our method makes to the structure of the generated logic, at line 533. In it, we point out that while most commonsense rules with many literals in their antecedent can be decomposed into more commonsensical sub-rules, it would be better if ARGOS were able to generate many-literal rules. Most commonsense rules with more than three atoms in the body of the clause (the antecedent) are often more commonsensical when decomposed into smaller sub-rules, meaning that commonsense is fully (and more naturally) described by only few-antecedent rules. For example, "Since it's winter and the fox is warm-blooded then to survive the winter the fox becomes white" can be described either as one long rule: $winter(season) \land must\textunderscore survive(fox) \land warm\textunderscore blooded(fox) \rightarrow white(fox)$, or as several sub-rules: (i) "Winter is cold" ($winter(season) \rightarrow cold(weather)$). (ii) "To survive, warm-blooded animals must stay warm" ($must\textunderscore survive(fox) \land warm\textunderscore blooded(fox) \rightarrow must\textunderscore stay\textunderscore warm(fox)$) (iii) To stay warm in the cold, the fox turns white ($cold(weather) \land must\textunderscore stay\textunderscore warm(fox) \rightarrow white(fox)$). Each of these sub-rules are more obvious/common-sensical than the longer 3-atom-head alternative.*
>
> ### Questions
>
> **Q1.** The authors should move the problem statement to be part of the background section, rather than having its own section. This would improve the paper's narrative flow.
>
> **Answer:**
> *Thank you, we have adopted this change in the updated version of the paper.*

---

> ### Author Response · Authors · 2025-11-27
>
> Hi, please let us know if we have answered your concerns to your satisfaction. If we have not, we ask that you please list your remaining concerns, so that we might address them

---

### Official Review · Reviewer_7mDQ · 2025-11-01

**Soundness:** 2
**Presentation:** 2
**Contribution:** 2
**Rating:** 4
**Confidence:** 4

**Summary:**

This paper addresses the problem of reasoning with missing commonsense information. It proposes a method of iteratively providing with an LLM the missing information in the form of L1 \land L2 \imply L3, where L1, L2, and L3 are all literals, and L1 and L2 are already deducible from the current premises. The paper experiments with 3 logical datasets and 3 less logical datasets. Experimental results demonstrate that the proposed method outperforms existing neural and neural-symbolic methods.

**Strengths:**

1.	The paper addresses the important problem of reasoning with missing commonsense information.

2.	The paper proposes a simple but potentially effective method to abduce and reasoning with the missing information. In contrast to neural-symbolic methods based on auto-formalization, the method resorts to more involved interaction of neural and symbolic methods.

3.	Experimental results demonstrate the viability of the proposed method.

**Weaknesses:**

1.	The paper states that it is dealing with abductive propositional logic problems (sec 4. Problem statement). But I believe the reasoning problem is first-order. Especially, the used dataset FOLIO is a typical dataset for natural language reasoning with first-order logic. The paper does not specify which SAT solver it is using.

2.	Some use of logical notions in the paper is improper. For example, Line 141: “Propositional logic is a logical system that involves propositions about variables”. This is not a proper introduction of propositional logic. Line 91: “contain variables not previously mentioned in the input problem“. I had difficulty understanding this in the beginning. But later, I understand it actually means “contain propositions”.

3.	Many logical notations used in the paper are problematic. I only give some examples here. The logic formula in Line 187 is incorrectly written. The formula in Line 214 is confusing. Proposition 1 in Appendix A is not well-stated.

4.	From Sec 6.2, it seems that the work of the paper is founded on perfect logical translation. On the one hand, auto-formalization is still a challenging topic. On the other hand, when the paper presents the performance of SAT-LM, I assume it is based on auto-formalization, then the comparison might be unfair.

**Questions:**

Does the paper deal with propositional or first-order reasoning? Which SAT solver is used?

---

> ### Author Response · Authors · 2025-11-21
> **Rebuttal to Initial Review**
>
> ### Weaknesses
>
> **w1.** The paper states that it is dealing with abductive propositional logic problems (sec 4. Problem statement). But I believe the reasoning problem is first-order. Especially, the used dataset FOLIO is a typical dataset for natural language reasoning with first-order logic.
>
> **Answer:**
> *FOLIO is originally a first-order logic dataset. In order to deal with abductive propositional logic problems, we modified it as described in Appendix F by having human annotators replace phrases with synonymous ones, introducing a missing proposition that $synonym_a \leftrightarrow synonym_b$. Also, in order to render it propositional (since we use a SAT solver), we unroll quantified expressions over all possible groundings. We have added text to this effect at line 338.*
>
> **w2.** The paper does not specify which SAT solver it is using.
>
> **Answer:**
> *We use the CADICAL solver. We have added this detail and a citation to the solver to the paper, at line 328.*
>
> **w3.** Some of the logical notions are used improperly in the paper. For example:
> - Line 141: "Prop. logic is a logical system that involves propositions about variables."
> - Line 91: "contain variables previously not mentioned in the input problem."
>
> **w4.** Many logical notations used are problematic. For example:
> - The formula at line 187 is incorrectly written.
> - The formula at line 214 is confusing.
> - Proposition 1 in Appendix A is not well-stated.
>
> *Thank you for bringing these to our attention. We have gone through the manuscript carefully. We have re-worked the problematic logical notations and improper use of logical notions. Please see the updated paper PDF for the changes, in which the changes are indicated by colored text. The relevant changes are in the Background section, in the Methodology section at lines 225 and 230, and Proposition 1 at line 708. We provide a more precise definition of propositional logic ("Propositonal logic is a logical system built around propositions, which are [...]", replace the notion of "variables not mentioned" with "propositions not instantiated", we re-write the definition of logical consistency as $P\land C \not \vdash \bot$, and have re-written the expression at line 225 to be clearer. We have also re-written Proposition 1 to be clearer and better-stated.*
>
> **w5.** From Sec 6.2, it seems that the work of the paper is founded on perfect logical translation. On the one hand, auto-formalization is still a challenging topic. On the other hand, when the paper presents the performance of SAT-LM, I assume it is based on auto-formalization, then the comparison might be unfair.
>
> **Answer:**
> *We thank the reviewer for raising this point. While we agree that the paper is founded on the assumption of perfect logical translation, we disagree that the comparison is unfair towards SAT-LM. SAT-LM is purely an auto-formalization method: it auto-formalizes the text input and then attempts to solve it via a first-order logic solver (Z3). By assuming perfect translation, we effectively assume zero error for SAT-LM. The fact that SAT-LM still struggles comes from the fact that it simply cannot make the necessary deductions with only the provided information, which is the point we aim to make by including it as a baseline. We have added new text at line 370 discussing this. With respect to the challenges of auto-formalization, we conduct an experiment in 5.2 to validate our claim that auto-formalization is a source of much less error than reasoning, finding that eliminating the assumption of perfect translation only marginally effects reasoning performance on FOLIO.*
>
> ### Questions
>
> **Q1.** Does the paper deal with propositional or first-order reasoning?
>
> **Answer:**
> *The paper deals with propositional reasoning. We state this in the Problem Setting at line 208, in which we state "We are given an abductive propositional logic problem [...]". However, we employ some first-order datasets, unrolling them into propositional logic by instantiating each potential grounding of the predicates as a separate propositional variable. We have added this detail at line 337 of the benchmarks description.*
>
> **Q2.** Which SAT solver is used?
>
> **Answer [repeated from W2]:**
> *We use the CADICAL solver. We have added this detail and a citation to the solver to the paper at line 328.*

---

> ### Author Response · Authors · 2025-11-27
>
> Hi, please let us know if we have answered your concerns to your satisfaction. If we have not, we ask that you please list your remaining concerns, so that we might address them

---

### Author Response · Authors · 2025-11-30
**Final Summary**

We thank the reviewers and the AC for their time and attention in their consideration of our work. The reviewers identified multiple important positive aspects of the work: important problem (7mDQ, L1Mk); novel, effective, insightful and intuitive method (7mDQ, mBHM, L1Mk); exceptionally clear (Kqo3); experimental methodology above reproach (Kqo3); strong performance compared to existing methods (mBHM); and ablation results demonstrating value of each contribution (mBHM). We summarize the reviewer's concerns, and our responses, below.

## Reviewer Concerns

1. **Assumption of perfect auto-formalization** (7mDQ, mBHM, and L1Mk). We provided additional empirical justification showing that performance is improved substantially even with imperfect auto-formalization. Reviewer L1Mk confirmed satisfaction with response and changes.

2. **Inconsistencies with propositional-logic literature** (7mDQ and L1Mk). All notation and descriptions were corrected to standard conventions, resolving L1Mk's concerns.

3. **Relevance given modern SOTA reasoning models (e.g., Gemini 2.5-pro)** (Kqo3). We clarified that improving open-source models remains scientifically important and revised the paper accordingly.

4. **Unbounded cost** (Kqo3). We clarified that the method is bounded and expanded the explanation in the paper.

5. **Limit of 3 literals in generated rules** (L1Mk and Kqo3). We justified the restriction via examples and the fact that any CNF can be transformed to 3-CNF; L1Mk was satisfied.

6. **Unrealistic task/data** (mBHM). We highlighted three realism-oriented datasets already included which the reviewer had overlooked.

7. **Lack of error analysis** (mBHM). We explained that the original appendix already contained such analysis.

8. **LLM-as-judge scoring reliability not evaluated** (mBHM) We added human-evaluated results confirming scoring-threshold reliability, but also highlighted that we do not use an LLM-as-judge to evaluate the performance of our method.

9. **Missing latency analysis** (mBHM) We pointed out latency metrics already reported in the appendix and clarified them further.

10. **Usefulness of added information** (mBHM) We added new empirical results showing ARGOS frequently contributes useful or proof-relevant information.

11. **General methodological rationale** (L1Mk) We provided additional explanation and empirical results, satisfying the reviewer.

**In conclusion, we believe that we have addressed all reviewer concerns, either empirically, through clarification, or by editing the text appropriately. Additionally, many of the general criticisms of the work were confirmed to have been addressed according to L1Mk both verbally and by an increase in score. L1Mk's diligence and attention to detail stands out among the reviewers, both in the initial review and in the discussion.**

---

### Meta-Review · Area_Chair_Q9oi · 2026-01-05

**Summary:**

This paper proposes an iterative scheme, named ARGOS, to improve LLMs' reasoning ability by augmenting a symbolic solver with new facts that are "abducted" by the LLM, until the problem can be solved up to a reasonable confidence. The authors show the effectiveness of ARGOS on several benchmarks for simple reasoning, removing common sense information and highlighting how ARGOS could retrieve it, or lead to the correct answer even when the first auto-formalization step can be imperfect.


The reviewers praised the simple and yet effective idea but at the same time criticized the paper for several imprecise statements, the lack of in-depth analysis of ARGOS shortcomings and the possibly limited applicability of the approach when it comes to very-large language models. The authors did a good rebuttal, addressing all the important concerns (see below), and this makes me think all reviewers would increase their scores to an average of 6.

As such the paper is accepted as poster.

**Reviewer Concerns:**

The first big concern is that the paper was rushed. This was clear in the presentation which contained a thin and partially wrong setting description, several inaccurate statements about the logic used and wrongly written logical statements were present. The authors recognized this and improved presentation a lot, taking into account all reviewers' suggestions. One minor comment on my side: make sure that footnotes go after punctuation and proof-read the whole paper, there are grammar typos in the orange part.

Another major concern was the assumption of perfect auto-formalization and the lack of a ablation to assess which sub-module is responsible for the largest error part (7mDQ, mBHM, and L1Mk). The authors addressed this to a somehow reasonable extent, providing a quick ablation experiment.

Reviewer Kqo3 raised some concerns (in a rather shallow review, I must say) about the problem significance w.r.t. frontier models. This is the only concern left open. To be honest, I agree with the authors and I don't think we should care much about closed-source models. Their performance cannot be quantified properly as we cannot know which tools or algorithms they are using nor performing any rigorous ablation on them. I also don't think that requiring access to logits (as raised by mBHM) is a strong limitation.

**Reviewer Scores:**

I believe all reviewers would have raised their scores as the main concerns have been addressed in the rebuttal. So going from 4 to perhaps 6, making the paper acceptable.

---

### Decision · Program_Chairs · 2026-01-26

Accept (Poster)